# HEEGNet: Hyperbolic Embeddings for EEG

**Shanglin Li**[1,2]**, Shiwen Chu**[1]**, Okan Koç**[2]**, Yi Ding**[4]**, Qibin Zhao**[2]**,**
**Motoaki Kawanabe**[1] **& Ziheng Chen**[3*]
[1] Advanced Telecommunications Research Institute International
[2] RIKEN Center for Advanced Intelligence Project
[3] University of Trento    [4] Nanyang Technological University

## ABSTRACT

Electroencephalography (EEG)-based brain-computer interfaces facilitate direct communication with a computer, enabling promising applications in human-computer interactions. However, their utility is currently limited because EEG decoding often suffers from poor generalization due to distribution shifts across domains (e.g., subjects). Learning robust representations that capture underlying task-relevant information would mitigate these shifts and improve generalization. One promising approach is to exploit the underlying hierarchical structure in EEG, as recent studies suggest that hierarchical cognitive processes, such as visual processing, can be encoded in EEG. While many decoding methods still rely on Euclidean embeddings, recent work has begun exploring hyperbolic geometry for EEG. Hyperbolic spaces, regarded as the continuous analogue of tree structures, provide a natural geometry for representing hierarchical data. In this study, we first empirically demonstrate that EEG data exhibit hyperbolicity and show that hyperbolic embeddings improve generalization. Motivated by these findings, we propose HEEGNet, a hybrid hyperbolic network architecture to capture the hierarchical structure in EEG and learn domain-invariant hyperbolic embeddings. To this end, HEEGNet combines both Euclidean and hyperbolic encoders and employs a novel coarse-to-fine domain adaptation strategy. Extensive experiments on multiple public EEG datasets, covering visual evoked potentials, emotion recognition, and intracranial EEG, demonstrate that HEEGNet achieves state-of-the-art performance. The code is available at `https://github.com/fightlesliefigt/HEEGNet`

## 1 INTRODUCTION

Electroencephalography (EEG) measures multi-channel electric brain activity (Niedermeyer & da Silva, 2005) and can reveal cognitive processes (Bell & Cuevas, 2012) and emotion states (Suhaimi et al., 2020). EEG-based brain-computer interfaces (BCI) aim to extract meaningful patterns for applications such as attention monitoring (Lee et al., 2015) and emotion recognition (Suhaimi et al., 2020). However, they currently suffer from poor generalization due to distribution shifts across sessions and subjects (Fairclough & Lotte, 2020).

In EEG-based neurotechnology, distribution shifts have traditionally been mitigated by collecting labeled calibration data and training domain-specific models (Lotte et al., 2018), which limits its utility and scalability (Wei et al., 2022). As an alternative, domain adaptation (DA) methods aim to learn a model from source domains that performs well on different (but related) target domains (Ben-David et al., 2010). In EEG, DA primarily addresses cross-session and cross-subject transfer learning problems (Wu et al., 2020). Since target domain data is typically unavailable during training and source domain data is not always available for privacy reasons, model adaptation is often treated in the context of multi-source multi-target source-free unsupervised domain adaptation (SFUDA) (Li et al., 2023). However, existing DA methods do not always work reliably, especially when the distribution shift between domains is large.

---

*Corresponding author.

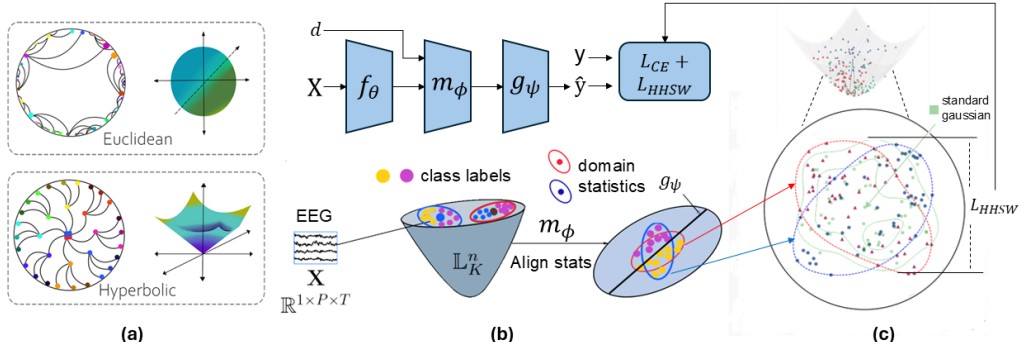

Figure 1: **Framework Overview.** (a) A comparison illustrating that hyperbolic space, with its negative curvature, is better suited for embedding hierarchical data than flat Euclidean space. (b) The HEEGNet architecture, which employs a coarse-to-fine domain adaptation strategy, DSMDBN (proposed). The first stage of DSMDBN aligns domain-specific moment statistics. (c) Top-down view in hyperbolic space: the second stage of DSMDBN aligns each source domain distribution to a standard hyperbolic Gaussian distribution by minimizing the Hyperbolic Horospherical Sliced-Wasserstein (HHSW) discrepancy.

Learning robust representations that capture underlying task-relevant information better could reduce distribution shifts and thereby improve generalization in domain adaptation (Bengio et al., 2013; Zhao et al., 2019). One promising approach to achieve such robustness is to exploit the underlying hierarchical structure in EEG data. Indeed, recent studies suggest that the brain's hierarchical cognitive processes, such as visual processing and emotion regulation, can be encoded in both intracranial EEG (Vlcek et al., 2020) and stimulus-evoked scalp EEG (Turner et al., 2023; Sun et al., 2023). While Euclidean embeddings currently dominate EEG decoding approaches, such embeddings are not well-suited to capture the exponential expansion of possible system states described by a hierarchical tree-like process (Peng et al., 2021). Intuitively, this is because, since the space is flat, the circumference and area of a circle grow only linearly and quadratically with the radius, respectively, leading to a mismatch with the underlying data geometry (Fig. 1a). In contrast, in hyperbolic spaces (Fig. 1a) with negative curvature, such quantities grow exponentially with the radius, thereby naturally approximating the exponential expansion of hierarchical processes (Krioukov et al., 2010). Leveraging this representational advantage, hyperbolic embeddings have outperformed Euclidean approaches across various tasks with hierarchical data in computer vision (CV) and natural language processing (NLP) (Ganea et al., 2018; Mettes et al., 2024).

In this study, we argue that hyperbolic spaces are often more appropriate for learning EEG embeddings. To this end, we first conduct a pilot study using the well-established EEGNet architecture (Lawhern et al., 2018) to generate EEG embeddings. We quantify the degree of hierarchical structure in these embeddings and confirm their hyperbolicity. We then modify EEGNet by replacing its multinomial logistic regression (MLR) with a hyperbolic variant, demonstrating that hyperbolic embeddings improve generalization over Euclidean ones. Motivated by the potential of the hyperbolic embeddings, we propose HEEGNet, a hybrid hyperbolic network architecture designed to capture the hierarchical structure of EEG data and learn domain-invariant hyperbolic EEG embeddings. HEEGNet (Fig. 3) integrates both Euclidean and hyperbolic encoders: the Euclidean encoders extract meaningful spectral–spatial–temporal EEG features and project them into hyperbolic space, with the hyperbolic encoders further refining these representations to capture hierarchical relationships more effectively. The hybrid nature of HEEGNet enables Euclidean encoders to leverage well-established signal processing principles and extract meaningful neurophysiological features from EEG signals, while the hyperbolic encoding helps to preserve the hierarchical structure in hyperbolic space. To address distribution shifts across domains, HEEGNet employs a novel *coarse-to-fine* domain alignment strategy. This strategy extends moment alignment, which is the current state-of-the-art (SotA) approach in EEG cross-domain generalization (Roy et al., 2022; Bakas et al., 2025) but whose performance often degrades under large distribution shifts (Li et al., 2025; Rodrigues et al., 2018). Specifically, we propose domain-specific moment-then-distribution batch normalization (DSMDBN, Sec. 3.1), which first explicitly aligns the domain-specific first- and second-order moments (Fig. 1b) using recently developed Riemannian batch normalization methods

(Chen et al., 2025a;b). In the second stage (Fig. 1c), the aligned EEG embeddings are further matched at the distribution level by minimizing the Hyperbolic Horospherical Sliced-Wasserstein discrepancy (Bonet et al., 2023) between each source domain and a standard hyperbolic Gaussian. DSMDBN transforms domain-specific inputs into domain-invariant outputs, enabling the extension to multi-source multi-target SFUDA scenarios. We demonstrate that HEEGNet obtains SotA performance on public EEG datasets encoding hierarchy, including visual evoked potentials, emotion recognition, and intracranial EEG. Additionally, we further validate its performance on public EEG motor imagery datasets, which are not reported to encode hierarchical information.

## 2 PRELIMINARIES

### 2.1 RELATED WORK

**Brain hierarchy.** The human brain is a complex network that supports various hierarchical cognitive processes. For example, in visual processing, lower cortical areas detect basic features, which higher cortical areas progressively refine into global representations (Hochstein & Ahissar, 2002). Similarly, in emotion regulation, subcortical areas generate primal urges, which are refined by the limbic system with experience, and the neocortex finally regulates them into complex thoughts and feelings (Panksepp, 2011). To study hierarchical brain dynamics, researchers have widely used intracranial EEG, which captures neuronal population activity with high spatial and temporal resolution (Lachaux et al., 2003). As for scalp EEG, Collins et al. (2018) observed distinct EEG responses to varied stimulus frequencies reflecting different visual processing levels. Sun et al. (2023) showed the hierarchical emotion ambiguity processing with distinct EEG patterns.

**Hyperbolic neural networks.** Hyperbolic neural networks, which perform neural network operations in hyperbolic space, have been widely explored in NLP and CV (Peng et al., 2021; Ganea et al., 2018). In the EEG literature, Chang et al. (2025) performed contrastive pretraining for emotion recognition in hyperbolic space. In contrast, our work introduces a hybrid hyperbolic network architecture and a domain adaptation strategy (DSMDBN), leveraging hyperbolic geometry to advance both representation learning and cross-domain generalization.

**Domain Adaptation in EEG.** Among the DA techniques applied to EEG, moments alignment that align the first and second-order moments either in input (He & Wu, 2019; Gnassounou et al., 2024) or in latent space (Kobler et al., 2022; Bakas et al., 2025) are considered as SotA (Roy et al., 2022; Bakas et al., 2025). Such heuristic alignments are not guaranteed to improve generalization (positive transfer), but tend to improve accuracy under mild distribution shifts (Yair et al., 2019). Theoretically, DA was studied as an upper bound analysis of target risk (Ben-David et al., 2010), using discrepancy terms (such as integral probability metrics (Redko et al., 2019) or f-divergences (Acuna et al., 2021)) between the source and target distributions, suggesting alignment of the feature distributions as a potential solution (Ganin et al., 2016). Inspired by this, the proposed DSMDBN (Fig. 1c) extends moment alignment further with feature distribution alignment.

### 2.2 MULTI-SOURCE MULTI-TARGET SFUDA

Let $\mathcal{X}$ denote the input space, $\mathcal{Y}$ the label space, and $\mathcal{D}$ the set of domain identifiers, where random variables $x$, $y$, and $d$ represent features, labels, and domains, respectively. In the multi-source multi-target SFUDA setting, given $M$ labeled source domain datasets $\mathcal{S} = \{(x_i, y_i, d_i) \mid x_i \in \mathcal{X}, y_i \in \mathcal{Y}, d_i \in \mathcal{D}_s\}_{i=1}^{L_s}$ and $N$ unlabeled target domain datasets $\mathcal{T} = \{(x_i, d_i) \mid x_i \in \mathcal{X}, d_i \in \mathcal{D}_t\}_{i=1}^{L_t}$, $L_s$ and $L_t$ denote the sizes of source domains and target domains, and $y_i$ and $d_i$ indicate the associated label and domain of sample $i$. We assume that all source and target domains share the same feature and label spaces. The goal is to learn a model $h$ from $\mathcal{S}$ that generalizes to unseen, unlabeled $\mathcal{T}$ via SFUDA, where only the trained model is available.

### 2.3 HYPERBOLIC GEOMETRY

The hyperbolic space is a Riemannian manifold of constant negative curvature $K < 0$. Among its five equivalent models (Cannon et al., 1997), we focus on the Lorentz model due to its numerical stability (Mishne et al., 2023). The $n$-dimensional Lorentz model is $\mathbb{L}_K^n := \{p \in \mathbb{R}^{n+1} \mid \langle p, p \rangle_{\mathcal{L}} = \frac{1}{K}, \ p_t > 0\}$ with the Lorentz inner product $\langle p, p \rangle_{\mathcal{L}} = \langle p_s, p_s \rangle - p_t^2$. Following Ratcliffe (2006),

we denote the first dimension $t$ as the time component and the remaining dimensions $s$ as the space component (i.e., $p \in \mathbb{L}_K^n = [p_t, p_s^\top]^\top$). The geodesic distance, defined as the shortest distance between any two points, is defined as:

$$d_{\mathcal{L}}(p, q) = \frac{1}{\sqrt{-K}} \cosh^{-1}\left(K \langle p, q \rangle_{\mathcal{L}}\right) \tag{1}$$

Under this distance, it is possible to introduce mean and variance operations into the Lorentz manifold. Specifically, for a set of points $P = \{p_i \in \mathbb{L}_K^n\}_{i \leq M}$, the weighted Fréchet mean $\text{wFM}_\eta$ on Riemannian manifolds is defined as the minimizer of the squared distances weighted by $\eta_i$:

$$\mu = \text{wFM}_\eta\left(\{p_i \in \mathbb{L}_K^n\}_{i=1}^M\right) = \arg \min_{q \in \mathbb{L}_K^n} \sum_{i=1}^M \eta_i \, d_{\mathcal{L}}^2(q, p_i), \tag{2}$$

If the weights $\eta_i$ are uniform, the weighted Fréchet mean reduces to the standard Fréchet mean, and the Fréchet variance $\nu^2$ is defined as the attained value at the Fréchet mean.

For each point $p \in \mathbb{L}_K^n$, there corresponds a tangent space $T_p\mathbb{L}_K^n = \{v \in \mathbb{R}^{n+1} | \langle p, v \rangle_{\mathcal{L}} = 0\}$. To project points between the manifold $p_i \in \mathbb{L}_K^n$ and the tangent space $v_i \in T_p\mathbb{L}_K^n$ at $p \in \mathbb{L}_K^n$, the exponential map $\exp_p^K : T_p\mathbb{L}_K^n \to \mathbb{L}_K^n$ and the logarithmic map $\log_p^K : \mathbb{L}_K^n \to T_p\mathbb{L}_K^n$ can be used to transport points $v_i \in T_p\mathbb{L}_K^n$ from the tangent space at $p$ to the tangent space at $q$, parallel transport $\text{PT}_{p \to q}(v) \in \mathbb{L}_K^n$ operation can be performed (see App. C.1 for their closed-form expressions).

Recently, Chen et al. (2025b, Sec. 5.5) show that Lorentz model admits a gyrovector structure (Chen et al., 2025b, Def. 4), which extends the vector addition and scalar multiplication into manifolds. Specifically, the Lorentz gyroaddition, gyromultiplication, and gyroinverse are

$$p \oplus_K^{\mathbb{L}} q = \text{Exp}_{\bar{0}}\left(\text{PT}_{\bar{0} \to p}\left(\text{Log}_{\bar{0}}(q)\right)\right), \quad \forall p, q \in \mathbb{L}_K^n, \tag{3}$$

$$t \odot_K^{\mathbb{L}} p = \text{Exp}_{\bar{0}}\left(t \, \text{Log}_{\bar{0}}(p)\right), \quad \forall t \in \mathbb{R}, \, \forall p \in \mathbb{L}_K^n, \tag{4}$$

$$\ominus_K^{\mathbb{L}} p = -1 \odot_K^{\mathbb{L}} p = \begin{bmatrix} p_t \\ -p_s \end{bmatrix}, \forall p \in \mathbb{L}_K^n, \tag{5}$$

where $\bar{0} = \left[\sqrt{-1/K}, 0, \ldots, 0\right]^\top$ is the origin over the Lorentz model. It is also the identity element: $\bar{0} \oplus_K^{\mathbb{L}} q = q, \forall q \in \mathbb{L}_K^n$. Eqs. (3) and (4) admit closed-form expressions, as shown in App. C.2.

### 2.3.1 HYPERBOLIC OPERATIONS.

**Hyperbolic neural networks.** We define the hyperbolic neural networks used in this work within the Lorentz model following Bdeir et al. (2024). App. B.1 provides a brief review of gyrovector spaces, which generalize vector structures to manifolds, serving as foundations to build neural networks in hyperbolic space. The Lorentz convolutional layer (App. C.6) is defined as a matrix multiplication between a linearized kernel and a concatenation of the values in its receptive field. The Lorentz ELU (App. C.3) activation applies the activation function to the space components and concatenates them with the time component. The average pooling layer is implemented by computing the Lorentzian weighted mean of all hyperbolic features within the receptive field. Analogous to the Euclidean MLR classifier, the Lorentz MLR (App. C.7) also utilizes the distance from instances to hyperplanes to describe the class regions.

**$\delta$-hyperbolicity.** Khrulkov et al. (2020) introduced $\delta$-hyperbolicity as a measure of the degree of tree-like structure inherent in embeddings. The idea is to find the smallest value of $\delta$ for which the triangle inequality holds via the Gromov product (mathematical formulation in App. C.8). In this formulation, the definition of a hyperbolic space in terms of the Gromov product can be interpreted as stating that the metric relations between any four points are the same as they would be in a tree, up to an additive constant $\delta$. The lower $\delta \geq 0$ is, the closer the embedding is to hyperbolic space.

**Hyperbolic horospherical sliced-Wasserstein discrepancy.** The sliced-Wasserstein distance (SWD) is a popular proxy for the Wasserstein distance for comparing probability distributions and has been extensively applied in optimal transport (Lee et al., 2019). Analogous to the Euclidean SWD, (Bonet et al., 2023) defined the hyperbolic sliced-Wasserstein distance by projecting distributions onto horospheres, denoted as Hyperbolic horospherical sliced-Wasserstein discrepancy

(HHSW), where distances between the projections of two points belonging to a geodesic with the same direction are conserved. For probability measures $\mu, \nu \in \mathcal{P}_p(\mathbb{L}^d)$ in the Lorentz model, with $p \geq 1$, the $p$-th power of the HHSW is defined as:

$$\text{HHSW}_p^p(\mu, \nu) = \int_{T_{\bar{0}}\mathbb{L}^d \cap S^d} W_p^p\left(B_{\#}^v \mu, B_{\#}^v \nu\right) d\lambda(v) \tag{6}$$

where $T_{\bar{0}}\mathbb{L}^d \cap S^d$ is the set of unit tangent vectors at the origin $\bar{0}$, $W_p^p$ is the $p$-th power of the 1D $p$-Wasserstein distance, $B_{\#}^v \mu$ denotes the horospherical projection of $\mu$ along direction $v$, and $d\lambda(v)$ is the uniform measure over these directions.

## 3 METHODS

### 3.1 DOMAIN-SPECIFIC MOMENT-THEN-DISTRIBUTION BATCH NORMALIZATION

**Hyperbolic batch normalization.** Batch normalization (BN) (Ioffe & Szegedy, 2015) is a widely used training technique in deep learning as BN layers speed up convergence and improve generalization. Chen et al. (2025a;b) extended the Euclidean BN into the hyperbolic space by gyrovector structure. The centering and scaling in the Euclidean BN correspond to Lorentz gyroaddition, gyroinverse, and gyromultiplication. Given a batch of activations $P = \{p_i \in \mathbb{L}_K^n\}_{i \leq M}$, the core operations of hyperbolic batch normalization (HBN) are

$$\text{HBN}(p_i) = \underbrace{\frac{\gamma}{\sqrt{\nu^2 + \epsilon}}}_{\text{Scaling}} \odot \underbrace{\left(\ominus \mu \oplus p_i\right)}_{\text{Centering}} \quad \forall i \leq M, \tag{7}$$

where $\mu$ and $\nu^2$ are Fréchet mean and Fréchet variance, $\gamma \in \mathbb{R}$ is the scaling parameter, and $\epsilon$ is a small value for numerical stability.

**Domain-specific momentum batch normalization for EEG.** Chang et al. (2019) introduced domain-specific batch normalization, which employs multiple parallel BN layers that process observations based on their corresponding domains to mitigate domain shift. However, in EEG scenarios, where small dataset sizes can make batch statistics unreliable for normalization (Yong et al., 2020). To address this, Kobler et al. (2022) proposed Domain-Specific Momentum Batch Normalization to track domain-specific momentum-based running estimates of the first- and second-order moments. It keeps two separate sets of running estimates, the training estimates are updated using a momentum parameter $\eta_{\text{train}(k)}$ that follows a clamped exponential decay schedule at training step $k$, while a fixed momentum parameter $\eta_{\text{test}}$ is used during testing.

**Domain-specific moment-then-distribution batch normalization.** While Kobler et al. (2022) achieved SotA performance, such moment alignment is not guaranteed to improve generalization (achieve positive transfer) and often struggles under large distribution shifts (Li et al., 2025; Rodrigues et al., 2018). On the other hand, feature distribution alignment offers theoretical guarantees, but can be challenging to achieve in practice, as the feature distributions may be too distant for effective learning (Chang et al., 2019). Thus, we propose a two-stage DSMDBN strategy by extending moment alignment in (Kobler et al., 2022) to incorporate the alignment of domain-specific feature distributions in hyperbolic space.

Formally, in our setting we assume that minibatches $\mathcal{B}_k$, that form the union of $N_{\mathcal{B}_k} \leq |\mathcal{D}|$ domain-specific minibatches $\mathcal{B}_k^d$, are drawn from distinct domains $d \in \mathcal{D}_{\mathcal{B}_k} \subseteq \mathcal{D}$. Each $\mathcal{B}_k^d$ contains $\frac{M}{N_{\mathcal{B}_k}}$ i.i.d. observations $x_i$, $i = 1, \ldots, M/N_{\mathcal{B}_k}$. In the first stage, $\text{DSMDBN}_{(1)}$ (algorithm 1) explicitly aligns the domain-specific running first- and second-order moments by centering and scaling them with $\nu_\phi^2$, which can be expressed as:

$$\tilde{p}_i = \text{DSMDBN}_{(1)}(p_i) = \text{HBN}^{d(i)}\left(p_i; \eta_{test}, \eta_{train(k)}\right). \quad \forall p_i \in \mathcal{B}_k^d, \quad \forall d \in \mathcal{D}_{\mathcal{B}_k} \tag{8}$$

$\text{DSMDBN}_{(2)}$ (algorithm 2), moment-aligned domain-specific EEG embeddings are then further (implicitly) aligned by matching them to samples from a standard hyperbolic Gaussian $\mathcal{N}(\bar{0}, 1)$. This matching is achieved by minimizing the HHSW (Eq. (6)) as a loss term

$$\text{DSMDBN}_{(2)}(p_i) = \text{HHSW}^{d(i)}(\tilde{p}_i), \quad \forall p_i \in \mathcal{B}_k^d, \quad \forall d \in \mathcal{D}_{\mathcal{B}_k}. \tag{9}$$

This implicit alignment guides the feature extractor toward learning robust, domain-invariant representations by matching source distributions to a standard Gaussian, thereby addressing distribution shifts that moment alignment alone often fails to mitigate. At test-time, HEEGNet applies only moment alignment, as source data are typically sufficiently diverse to capture the task-relevant variability in EEG (Rodrigues et al., 2018; Mellot et al., 2023). Learning a Gaussian-aligned feature space from the sources is sufficient for the extractor to produce normalized target-domain features after moment alignment, enabling the application of domain matching in the SFUDA scenarios.

### 3.2 HEEGNET

Following Kobler et al. (2022), we constrain the hypothesis class $\mathcal{H}$ to functions $h : \mathcal{X} \times \mathcal{D} \to \mathcal{Y}$ that can be decomposed into a composition of a shared feature extractor $f_\theta : \mathcal{X} \to \mathbb{L}_K^n$, a domain-specific alignment module $m_\phi : \mathbb{L}_K^n \times \mathcal{D} \to \mathbb{L}_K^n$, and a shared classifier $g_\psi : \mathbb{L}_K^n \to \mathcal{Y}$ with parameters $\Theta = \{\theta, \phi, \psi\}$. We parametrize $h = g_\psi \circ m_\phi \circ f_\theta$ as a neural network and learn the entire model in an end-to-end fashion, which we denote as HEEGNet (details in App. D.2).

The HEEGNet is designed as a hybrid model, combining Euclidean encoders with hyperbolic neural networks. Most existing hyperbolic models adopt a hybrid approach, first generating hierarchical embeddings in Euclidean space and then mapping them to hyperbolic space, leveraging the strong feature extraction capabilities of well-established Euclidean encoders (Peng et al., 2021). This hybrid design is particularly well-suited for EEG data. Studies suggest that EEG signals contain hierarchical information across temporal (Damera et al., 2020), spectral (Sun et al., 2023), and spatial (Tonoyan et al., 2017) dimensions. Euclidean encoders, such as convolutional networks, naturally align with this organization, making them effective for extracting hierarchical and neurophysiologically meaningful features directly from raw EEG signals (Lawhern et al., 2018). However, such operations do not preserve physical properties in hyperbolic space, where dimensions are intrinsically coupled (see App. F for fully hyperbolic experiments).

In a nutshell, the feature extractor $f_\theta$ consists of three Euclidean convolutional layers along with standard components (e.g., BN and pooling), a projection layer ProjX, and a hyperbolic convolutional layer (Fig. 3). We sequentially adopt the three convolutional layers from EEGNet (Lawhern et al., 2018): temporal convolution to learn frequency-specific filters, depthwise spatial convolution to capture electrode-wise patterns, and a second depthwise temporal convolution to summarize information across time. These well-established operations provide spectral-spatial-temporal feature maps with meaningful neurophysiologically properties. The ProjX layer projects the Euclidean feature maps into hyperbolic space $\mathbb{L}_K^n$, after which a hyperbolic convolutional layer performs pointwise convolution to optimally combine these feature maps. The alignment module $m_\phi$ applies the first stage of the proposed DSMDBN (Eq. (8)) to explicitly align domain-specific moments in hyperbolic space, followed by hyperbolic ELU and pooling (Eq. (21)) for dimension reduction. Finally, the classifier $g_\psi$ is parametrized as a hyperbolic MLR (HMLR) layer).

## 4 EXPERIMENTS

In this study, we consider three EEG modalities that have been reported to encode hierarchical structures: visual- and emotion-stimuli scalp EEG, and intracranial EEG (Turner et al., 2023; Sun et al., 2023; Sonkusare et al., 2020). Corresponding EEG-based BCI applications include steady-state visually evoked potentials (SSVEP), code-modulated visually evoked potentials (CVEP), emotion recognition, and intracranial EEG. While all applications hold significant potential for rehabilitation and healthcare (Al-Nafjan et al., 2017; Guo et al., 2022; Elsner et al., 2018), their practical utility remains limited due to poor generalization across domains. We conduct a pilot study to evaluate hyperbolic embeddings and perform comprehensive experiments to assess the proposed HEEGNet. We further evaluate HEEGNet on public motor imagery datasets that are not being reported to encode hierarchical information.

**Visually evoked potentials (VEP)**: *Nakanishi* (9 subjects/1 session/12 classes) (Nakanishi et al., 2015), *Wang* (34/1/40) (Wang et al., 2016), *CBVEP40* (12/1/4) (Castillos et al., 2023), and *CB-VEP100* (12/1/4) (Castillos et al., 2023). We used MOABB (Chevallier et al., 2024) to pre-process the data and extract labeled epochs. Following Pan et al. (2022b), EEG signals were resampled to 256 Hz, bandpass filtered between 1-50 Hz, and segmented each trial into 1- or 2-second segments.

**Emotion recognition**: *Seed* (15/3/3) (Duan et al., 2013) and *Faced* (123/1/9) (Chen et al., 2023b). We used the public available pre-processed data. For Seed, EEG signals were resampled to 200 Hz and bandpass filtered between 0–75 Hz; for Faced, a 0.05–47 Hz bandpass filter was applied.

**Intracranial EEG**: *Boran* (9/2-7/2) (Boran et al., 2020). We use MNE-python (Gramfort et al., 2014) to pre-process data and extract labeled epochs. Following Frauscher et al. (2018), EEG signals were resampled to 1000 Hz, bandpass filtered between 0.3-70 Hz.

**Evaluation.** We consider cross-domain adaptation within each dataset. We treat sessions as domains, and use either a leave-one-group-out (source domain number $\leq 10$) or a 10-fold leave-groups-out cross-validation scheme to fit and evaluate models. For the intracranial EEG dataset, due to the different number of electrodes across subjects, we consider the cross-session adaptation setting. We fit and evaluate models independently for each subject, treating the session as the grouping variable. For other datasets, we consider the cross-subject adaptation and treat the subject as the grouping variable. We use either the standard Adam optimizer (Kingma & Ba, 2014) for Euclidean frameworks or the Riemannian Adam optimizer (Bécigneul & Ganea, 2018) for geometric frameworks, both with default hyperparameters in PyTorch. We split the source domain data into training and validation sets (80% / 20% splits, randomized and stratified by domain and label) and iterated through the training set for 100 epochs. Early stopping was fit with a single stratified (domain and labels) inner train/validation split.

## 4.1 PILOT STUDY.

We select five datasets as representatives to motivate the use of hyperbolic embeddings. All experiments are conducted under the cross-domain setting and follow the evaluation scheme described above. We begin by training and evaluating EEGNet on cross-domain tasks. Following Krioukov et al. (2010), we then use the target domain raw EEG data and the trained EEGNet to generate embeddings of the intermediate layer (after first two convolutional layers), and the classification space to quantify their degree of inherent tree-likeness using $\delta$-hyperbolicity respectively (Eq. (27)). Following Khrulkov et al. (2020), we report the scale-invariant metric $\delta_{rel} \in [0, 1]$, where the lower $\delta_{rel}$ is the higher the hyperbolicity of the embeddings. We then modify EEGNet by replacing its MLR with its hyperbolic variant, HMLR, and repeat the training and evaluation procedure.

Tab. 1 indicates that all dataset both raw EEG data and EEGNet generated embeddings exhibit hierarchical structures, confirming the suitability of hyperbolic geometry for capturing the underlying information. Furthermore, replacing the Euclidean MLR layer in EEGNet with its hyperbolic counterpart consistently enhances performance across all datasets (Tab. 2), suggesting that hyperbolic geometry captures more robust representations across domains and improves generalization. t-SNE visualizations Fig. 2a, b for the identical subject and session show that under the same encoder structure, hyperbolic representations enhance class separability. These findings strongly motivate the use of hyperbolic embeddings in cross-domain generalization.

Table 1: **Pilot Study:** $\delta_{rel}$ **of datasets**. The lower the $\delta_{rel}$, the closer the dataset to hyperbolic space. The number of domains in each dataset is indicated by $n$.

| | Visual | | Emotion | | Intracranial |
|---|---|---|---|---|---|
| **Dataset** | Nakanishi | Wang | Seed | Faced | Boran |
| **Metric** | (n=9) | (n=34) | (n=45) | (n=123) | (n=37) |
| Raw $\delta_{rel}$ | $0.244 \pm 0.064$ | $0.219 \pm 0.053$ | $0.052 \pm 0.026$ | $0.103 \pm 0.077$ | $0.157 \pm 0.045$ |
| Intermediate $\delta_{rel}$ | $0.263 \pm 0.036$ | $0.240 \pm 0.045$ | $0.061 \pm 0.026$ | $0.111 \pm 0.050$ | $0.141 \pm 0.041$ |
| Classification space $\delta_{rel}$ | $0.306 \pm 0.027$ | $0.333 \pm 0.039$ | $0.072 \pm 0.025$ | $0.132 \pm 0.047$ | $0.017 \pm 0.048$ |

## 4.2 MAIN EXPERIMENTS

**Baseline models.** We included six deep learning architectures EEGNet (Lawhern et al., 2018), EEGConformer (Song et al., 2022), ATCNet (Altaheri et al., 2022), TSLANet (Eldele et al., 2024), SchirrmeisterNet (Schirrmeister et al., 2017), and FBCNet (Mane et al., 2021) that proposed or

Table 2: **Pilot Study: comparison of Euclidean / Hyperbolic MLR**. The averages of test-set scores are shown above using balanced accuracy (the score and the standard deviation are shown for each dataset). The number of domains in each dataset is indicated by $n$. Permutation-paired t-tests were used to identify significant differences between EEGNet+HMLR (*Hyperbolic*) and EEGNet (1e4 permutations). Significant differences are highlighted using dots: $\cdot p \leq 0.05$, $\bullet p \leq 0.01$, $\bullet p \leq 0.001$.

| | Visual | | Emotion | | Intracranial |
|---|---|---|---|---|---|
| **Dataset** | Nakanishi | Wang | Seed | Faced | Boran |
| **Model** | $(n = 9)$ | $(n = 34)$ | $(n = 45)$ | $(n = 123)$ | $(n = 37)$ |
| EEGNet | $57.2 \pm 19.8$ • | $37.4 \pm 12.9$ • | $74.5 \pm 19.8$ | $24.8 \pm 13.7$ • | $55.4 \pm 9.1$ |
| EEGNet + HMLR | $60.8 \pm 20.7$ | $39.2 \pm 13.8$ | $75.1 \pm 20.0$ | $38.8 \pm 12.4$ | $57.4 \pm 8.7$ |

extensively used for general EEG decoding. We consider four deep learning architectures specifically designed for VEP: EEGInception (Santamaria-Vazquez et al., 2020), DDGCNN (Zhang et al., 2024), SSVEPNet (Pan et al., 2022a), SSVEPFormer (Chen et al., 2023a); and two deep learning architectures for emotion recognition: EmT (Ding et al., 2025), TSception (Ding et al., 2022). We further considered two foundation models: LaBraM (Jiang et al., 2024) and CBraMod (Wang et al., 2024a) in evaluation. We use the implementation provided in the public available repositories for all architectures, stick to all hyperparameters as provided, and use the standard cross-entropy loss as training objective.

**SFUDA baselines.** We consider two SFUDA baselines: Euclidean alignment (EA) (He & Wu, 2019), and spatio-temporal Monge alignment (STMA) (Gnassounou et al., 2024). These alignment methods are model-agnostic techniques that are applied to the EEG data in the input space before a model is fitted. For example, EA aligns the EEG trials covariance matrix directly in the input space. Therefore, we combine them with different models (e.g., EEGNet + EA) in our evaluation.

**HEEGNet.** We parametrize $h = g_\psi \circ m_\phi \circ f_\theta$ as a neural network and learn the entire model in an end-to-end fashion, and use the standard cross-entropy loss with HHSW (Eq. (9)) as the training objective. The HHSW loss weight is a hyperparameter to be tuned, we set it to 0.01 for emotion and 0.5 for other datasets in our experiments.

Following Bdeir et al. (2024), we set the curvature to -1 by default and implemented the norm normalization to assure numerical stabilization for training. Computational cost for hyperbolic operation is briefly discussed in App. D.4. During source-free target domain adaptation, HEEGNet keeps the fitted source feature extractor $f_\theta$ and linear classifier $g_\psi$ fixed and estimates domain-specific first- and second-order statistics by solving Eq. (2) for moments alignment $m_\phi$.

Tab. 3 summarizes the results across all three EEG modalities, presenting the grand average scores with general EEG decoding baseline methods. Extended results for VEP-specific, emotion recognition-specific and foundation models, as well as per-dataset results, are provided in Supplementary Tab. 8 and Tab. 9. At the overall grand average level, HEEGNet (DSMDBN) outperforms all baseline methods. A visualization of the two stages of DSMDBN is shown in Fig. 2c, d. Interestingly, HEEGNet (DSMDBN+EA), the integration of DSMDBN with the input space alignment method EA, achieves superior performance. We conduct an ablation study to systematically investigate the effects of different alignment strategies within the HEEGNet architecture.

**Ablation study.** Tab. 4 summarizes the effectiveness of three different alignment methods, input alignment (EA), moment alignment, and distribution alignment, within the HEEGNet architecture. We highlight three important observations. First, moment alignment in hyperbolic space is the primary driver of performance improvement. The absence of moment alignment consistently results in a significant performance drop of at least 12.3% compared to the best-performing configuration. Second, distribution alignment proves effective only when paired with moment alignment, validating our proposed DSMDBN approach, which first aligns moments to facilitate subsequent distribution alignment. Third, moment alignment in the latent space outperforms input space alignment, supporting the findings of Bakas et al. (2025) that alignment benefits from enhanced class discrimination in the latent representation. Fourth, DSMDBN augmented with input alignment yields the best performance, indicating that multistage alignment a promising strategy in cross-domain generalization.

Table 3: **Main experiment results.** Grand average of test-set scores across three EEG modalities (balanced accuracy (%); higher is better; mean ± std) Permutation-paired t-tests were used to identify significant differences between HEEGNet (DSMDBN+EA) and baseline methods (1e4 permutations, 18 tests, t-max correction). Significance markers: $\cdot p \leq 0.05$, $\bullet p \leq 0.01$, $\bullet p \leq 0.001$.

| Model | SFUDA | VEP (n=67) | Emotion (n=168) | Intracranial (n=37) | Overall (n=272) |
|---|---|---|---|---|---|
| EEGNet | w/o | 35.7±15.5 ● | 38.1±27.0 ● | 55.4±9.1 ● | 39.9±23.6 ● |
| | EA | 55.8±20.8 ● | 73.8±22.3 ● | 59.1±11.2 ● | 67.4±22.3 ● |
| | STMA | 30.3±8.9 ● | 70.7±20.5 ● | 51.2±5.9 ● | 58.1±24.0 ● |
| EEGConformer | w/o | 29.1±9.1 ● | 83.2±16.9 · | 55.5±7.0 ● | 66.1±27.2 ● |
| | EA | 46.1±23.1● | 82.8±16.8 ● | 61.7±13.1 · | 70.9±24.1 ● |
| | STMA | 27.8±7.0 ● | 78.2±15.9 ● | 53.1±5.2 ● | 62.4±25.2 ● |
| ATCNet | w/o | 33.3±15.4 ● | 17.5±11.7 ● | 57.5±9.0 ● | 26.9±18.5 ● |
| | EA | 52.2±21.4● | 54.1±15.5 ● | 57.1±9.0 ● | 54.0±16.5 ● |
| | STMA | 39.4±15.6 ● | 56.4±16.3 ● | 54.2±5.7 ● | 51.9±16.7 ● |
| TSLANet | w/o | 24.4±14.1 ● | 35.5±12.9 ● | 55.1±9.3 ● | 35.5±15.7 ● |
| | EA | 37.3±28.1 ● | 44.8±18.1 ● | 54.9±10.4 ● | 44.3±20.8 ● |
| | STMA | 20.5±5.4 ● | 49.6±15.1 ● | 54.1±6.2 ● | 43.0±17.9 ● |
| SchirrmeisterNet | w/o | 40.2±27.5 ● | 51.9±22.4 ● | 56.9±7.7 ● | 49.7±23.1 ● |
| | EA | 35.7±24.7 ● | 79.0±16.4 ● | 65.0±10.2 | 66.5±25.7 ● |
| | STMA | 33.1±16.8 ● | 64.2±16.7 ● | 53.9±5.5 ● | 55.2±20.4 ● |
| FBCNet | w/o | 22.8±8.0 ● | 28.5±19.5 ● | 61.1±10.6 · | 31.6±20.2 ● |
| | EA | 38.8±28.4 ● | 51.3±20.7 ● | 61.3±11.0 · | 49.6±22.9 ● |
| | STMA | 21.6±5.4 ● | 33.0±20.9 ● | 53.0±6.8 ● | 32.9±19.2 ● |
| HEEGNet *(proposed)* | DSMDBN | **79.6±19.8** | 82.8±16.6 | 58.4±9.6 | 78.7±18.6 |
| | DSMDBN+EA | 77.4±12.6 | **86.7±12.5** | **66.7±12.5** | **81.7±14.4** |

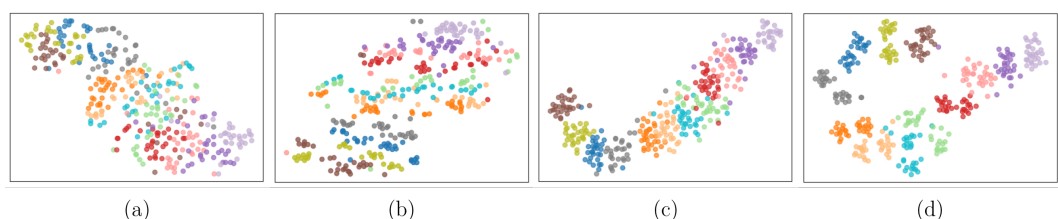

| (a) | (b) | (c) | (d) |

Figure 2: **t-SNE visualizations of classification space for Nakanishi (subject 4, session1).** Each color denotes a distinct class. (a) EEGNet, (b) EEGNet with HMLR, (c) DSMDBN stage 1, and (d) DSMDBN stage 2. The plots illustrate how hyperbolic embedding and our proposed two-stage alignment progressively enhance class separability.

### 4.3 MOTOR IMAGERY

We consider two publicly available motor imagery datasets. Pre-processing included resampling EEG signals to 250 or 256 Hz, applying temporal filters to capture frequencies between 4 and 36 Hz, and extracting 3-second epochs linked to specific class labels. In addition to Euclidean models, we evaluate several manifold-based decoding approaches for motor imagery. These include Grassmann manifold methods: GDLNet (Wang et al., 2024b) and GyroAtt-Gr (Wang et al., 2025); Symmetric Positive Definite (SPD) manifold methods: TSMNet (Kobler et al., 2022), MAtt (Pan et al., 2022b), and CSPNet (Ju & Guan, 2022); as well as Symmetric Positive Semi-Definite (SPSD) manifold methods: GyroAtt-SPSD (Wang et al., 2025). The results, summarized in Tab. 5, indicate that our proposed HEEGNet delivers competitive performance.

Table 4: **Ablation results.** Grand average of all test-sets scores (balanced accuracy (%), higher is better) relative to the combination of alignment in HEEGNet. Permutation-paired t-test values and adjusted p-values indicate the effect strength (1e4 permutations, 7 tests, t-max correction).

| Alignment | | | Metrics | |
|---|---|---|---|---|
| Moments | Distribution | Input | mean (std) | t-val (p-val) |
| ✓ | ✓ | ✓ | - | - |
| ✓ | ✓ | ✗ | -3.0 (18.6) | 3.6 (0.0003) |
| ✓ | ✗ | ✓ | -5.0 (15.6) | 8.8 (0.0001) |
| ✓ | ✗ | ✗ | -5.0 (19.3) | 5.5 (0.0001) |
| ✗ | ✓ | ✗ | -12.3 (23.1) | 10.4 (0.0001) |
| ✗ | ✗ | ✗ | -12.9 (22.4) | 11.7 (0.0001) |
| ✗ | ✓ | ✓ | -15.5 (21.4) | 14.9 (0.0001) |
| ✗ | ✗ | ✓ | -15.5 (22.2) | 16.0 (0.0001) |

Table 5: **Motor imagery per dataset results.** Average of test-set scores (balanced accuracy (%); higher is better; mean ± std).

| | | | Dataset | |
|---|---|---|---|---|
| **Manifold** | **Model** | **SFUDA** | **BNCI2014001** | **BNCI2015001** |
| Euclidean | ATCNet | w/o | $42.7 \pm 16.4$ | $60.2 \pm 8.4$ |
| | EEGConformer | w/o | $42.6 \pm 16.7$ | $60.1 \pm 10.7$ |
| | EEGNet | w/o | $43.6 \pm 16.7$ | $61.3 \pm 8.8$ |
| | | EA | $49.9 \pm 16.9$ | $72.5 \pm 14.2$ |
| | | STMA | $49.7 \pm 16.9$ | $69.9 \pm 14.6$ |
| | EEGInceptionMI | w/o | $39.7 \pm 12.7$ | $59.5 \pm 9.2$ |
| | ShallowNet | w/o | $42.2 \pm 16.2$ | $58.7 \pm 5.8$ |
| | LaBraM | w/o | $33.3 \pm 16.6$ | $70.8 \pm 31.4$ |
| | CBraMod | w/o | $30.6 \pm 4.0$ | $59.1 \pm 8.1$ |
| Grassmann | GDLNet | w/o | $46.3 \pm 5.1$ | $63.3 \pm 14.2$ |
| | GyroAtt-Gr | w/o | $52.1 \pm 14.2$ | $75.3 \pm 13.7$ |
| SPSD | GyroAtt-SPSD | w/o | $51.7 \pm 13.1$ | $74.9 \pm 12.6$ |
| SPD | MAtt | w/o | $45.3 \pm 11.3$ | $63.1 \pm 10.1$ |
| | CSPNet | w/o | $45.2 \pm 9.3$ | $64.2 \pm 13.4$ |
| | TSMNet | w/o | $43.0 \pm 13.3$ | $61.7 \pm 11.4$ |
| | | EA | $51.2 \pm 15.1$ | $72.5 \pm 13.6$ |
| | | STMA | $52.5 \pm 16.4$ | $70.1 \pm 14.2$ |
| | | SPDDSBN | $\underline{54.6 \pm 16.1}$ | $74.3 \pm 14.7$ |
| Hyperbolic | HEEGNet | DSMDBN+EA | $54.1 \pm 15.9$ | $\underline{\mathbf{75.8 \pm 13.0}}$ |

## 5 DISCUSSIONS

In this work, we introduced HEEGNet, a hybrid hyperbolic network architecture designed to capture the hierarchical structure of EEG data and learn domain-invariant hyperbolic embeddings. Our pilot study and empirical analyses indicate that EEG data exhibits hyperbolicity and that hyperbolic embeddings improve generalization compared to Euclidean ones. By integrating both Euclidean and hyperbolic encoders and employing a novel two-stage domain adaptation strategy (DSMDBN), HEEGNet effectively aligns domain-specific moments and distributions in hyperbolic space. Extensive experiments demonstrate state-of-the-art performance. Despite these strengths, like all hyperbolic neural networks, hyperbolic operations add additional computational cost. Future work will explore online extensions of hyperbolic normalization and the development of more expressive encoders capable of capturing hierarchical structure in EEG signals.

## 6 ACKNOWLEDGMENTS

This project was supported by the JSPS Bilateral Program under Grant Number JPJSBP120257420.

Motoaki Kawanabe was partially supported by the Innovative Science and Technology Initiative for Security Grant Number JPJ004596, ATLA, Japan.

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

APPENDIX CONTENTS

## A  LARGE LANGUAGE MODEL USAGE STATEMENT

Large language models were partially used in this article to refine the contents.

## B  GYROVECTOR SPACES

### B.1  GENERAL DEFINITION

This subsection briefly reviews the gyrovector space (Ungar, 2022), which generalizes the vector structure into manifolds. It has shown great success in building hyperbolic neural networks (Ganea et al., 2018; Chami et al., 2019; Shimizu et al., 2021).

We start from the gyrogroup. Intuitively, gyrogroups are natural generalizations of groups. Unlike groups, gyrogroups are non-associative but have gyroassociativity characterized by gyrations.

**Definition B.1** (Gyrogroups (Ungar, 2022)). Given a nonempty set $G$ with a binary operation $\oplus :
G \times G \rightarrow G$, $(G, \oplus)$ forms a gyrogroup if its binary operation satisfies the following axioms for any $p, q, z \in G$ :

(G1) There is at least one element $e \in G$ called a left identity (or neutral element) such that $e \oplus p = p$.

(G2) There is an element $\ominus p \in G$ called a left inverse of $p$ such that $\ominus p \oplus p = e$.

(G3) There is an automorphism $\mathrm{gyr}[p, q] : G \rightarrow G$ for each $p, q \in G$ such that

$$p \oplus (q \oplus z) = (p \oplus q) \oplus \mathrm{gyr}[p, q]z \quad \text{(Left Gyroassociative Law)}.$$

The automorphism $\mathrm{gyr}[p, q]$ is called the gyroautomorphism, or the gyration of $G$ generated by $p, q$.

(G4) Left reduction law: $\mathrm{gyr}[p, q] = \mathrm{gyr}[p \oplus q, q]$.

**Definition B.2** (Gyrocommutative Gyrogroups (Ungar, 2022)). A gyrogroup $(G, \oplus)$ is gyrocommutative if it satisfies

$$p \oplus q = \mathrm{gyr}[p, q](q \oplus p) \quad \text{(Gyrocommutative Law)}.$$

Similarly, the gyrovector space generalizes the vector space,

**Definition B.3** (Gyrovector Spaces (Ungar, 2022; Chen et al., 2025b)). A gyrocommutative gyrogroup $(G, \oplus)$ equipped with a scalar gyromultiplication $\otimes : \mathbb{R} \times G \rightarrow G$ is called a gyrovector space if it satisfies the following axioms for $s, t \in \mathbb{R}$ and $p, q, z \in G$:

(V1) Identity Scalar Multiplication: $1 \otimes p = p$.

(V2) Scalar Distributive Law: $(s + t) \otimes p = s \otimes p \oplus t \otimes p$.

(V3) Scalar Associative Law: $(st) \otimes p = s \otimes (t \otimes p)$.

(V4) Gyroautomorphism: $\mathrm{gyr}[p, q](t \otimes z) = t \otimes \mathrm{gyr}[p, q]z$.

(V5) Identity Gyroautomorphism: $\mathrm{gyr}[s \otimes p, t \otimes p] = \mathbb{I}$, where $\mathbb{I}$ is the identity map.

The vector space, equipped with addition and scalar multiplication, forms the foundation of Euclidean deep learning. Similarly, the gyrovector space, endowed with gyroaddition and scalar gyromultiplication, offers a powerful tool for designing neural networks over non-Euclidean manifolds.

## C   LORENTZ OPERATIONS

### C.1   RIEMANNIAN OPERATORS

The exponential map $\text{Exp}_p^K : T_p\mathbb{L}_K^n \to \mathbb{L}_K^n$ and logarithmic map $\text{Log}_p^K : T_p\mathbb{L}_K^n \to \mathbb{L}_K^n$ project points between the manifold $p_i \in \mathbb{L}_K^n$ and the tangent space $v_i \in T_p\mathbb{L}_K^n$ at point $p \in \mathbb{L}_K^n$.

$$\text{Exp}_p^K(v) = \cosh(\alpha)p + \sinh(\alpha)\frac{v}{\alpha}, \quad \text{with} \quad \alpha = \sqrt{-K}\,\|v\|_{\mathcal{L}}, \quad \|v\|_{\mathcal{L}} = \sqrt{\langle v, v \rangle_{\mathcal{L}}} \tag{10}$$

$$\text{Log}_p^K(q) = \frac{\cosh^{-1}(\beta)}{\sqrt{\beta^2 - 1}} \cdot (q - \beta p), \quad \text{with} \quad \beta = K\langle p, q \rangle_{\mathcal{L}} \tag{11}$$

To transport points $v_i \in T_p\mathbb{L}_K^n$ from the tangent space at $p$ to the tangent space at $q$, parallel transport $\text{PT}_{p \to q}(v) \in \mathbb{L}_K^n$ can be used:

$$\text{PT}_{p \to q}(v) = v - \frac{K(q, v)_K}{1 + K(p, q)_K}(p + q) \tag{12}$$

### C.2   GYROVECTOR OPERATORS

The definitions in Eqs. (3) and (4) are direct generalizations of Euclidean vector operations. In Euclidean space, vector addition and scalar multiplication can be understood geometrically as operations on rays emanating from the origin.

Let us first review Euclidean geometry. For $\mathbb{R}^n$, we have

$$T_p\mathbb{R}^n = T_q\mathbb{R}^n = \mathbb{R}^n, \tag{13}$$
$$\text{Log}_p(q) = q - p, \tag{14}$$
$$\text{Exp}_p(v) = v + p \tag{15}$$
$$\text{PT}_{p \to q}(v) = v, \tag{16}$$

where $p, q \in \mathbb{R}^n$ and $v \in T_x\mathbb{R}^n$

To compute the Euclidean vector addition $p + q$, one may regard $q$ as the ray from $\mathbf{0}$ to $q$, parallel translate this ray to the base point $p$, and then shoot it out from $p$. The above process can be expressed as

$$p + q = \text{Exp}_p\big(\text{PT}_{\mathbf{0} \to p}(\text{Log}_{\mathbf{0}}(q))\big). \tag{17}$$

Similarly, Euclidean scalar multiplication corresponds to taking the ray from $\mathbf{0}$ to $p$, scaling its length by $t$, and then shooting it out from the origin again:

$$t \odot p = \text{Exp}_{\mathbf{0}}(t\,\text{Log}_{\mathbf{0}}(p)) = tp. \tag{18}$$

The Lorentz gyroaddition and gyromultiplication extend this geometric intuition of Euclidean linear operations to curved manifolds.

**Expressions.** Let $p = [p_t, p_s]^\top$ and $q = [q_t, q_s]^\top$ be points in $\mathbb{L}_K^n$, where $p_t, q_t \in \mathbb{R}$ are the time scalars, and $p_s, q_s \in \mathbb{R}^n$ are the spatial parts. Let $t \in \mathbb{R}$ be a real scalar. The following reviews the closed-form expression of the Lorentz gyro operators, which are more efficient than the Riemannian definitions Eqs. (3) and (4) (Chen et al., 2025b, Sec. 6.1). The Lorentz gyroaddition

and gyromultiplication have the closed-form solution:

$$
\text{Gyroaddition:} \quad p \oplus_K^{\mathcal{M}} q = \begin{cases} p, & q = \bar{0}, \\ q, & p = \bar{0}, \\ \begin{bmatrix} 1 - \dfrac{D - KN}{|K|} \\ \dfrac{D + KN}{2(A_s p_s + A_q q_s)} \\ \dfrac{}{D + KN} \end{bmatrix}, & \text{Others.} \end{cases}
\tag{19}
$$

$$
\text{Gyromultiplication:} \quad t \otimes_K^{\mathcal{M}} p = \begin{cases} \bar{0}, & t = 0 \ \vee \ p = \bar{0}, \\ \dfrac{1}{\sqrt{|K|}} \begin{bmatrix} \cosh\left(t \cosh^{-1}(\sqrt{|K|} p_t)\right) \\ \dfrac{\sinh\left(t \cosh^{-1}(\sqrt{|K|} p_t)\right)}{\|p_s\|} p_s \end{bmatrix}, & t \neq 0, \end{cases}
\tag{20}
$$

where $A_s = ab^2 - 2Kbs_{pq} - Kan_q$ and $A_q = b(a^2 + Kn_p)$ with the following:

$$
a = 1 + \sqrt{|K|} p_t, \quad b = 1 + \sqrt{|K|} q_t, \quad n_p = \|p_s\|^2, \quad n_q = \|q_s\|^2, \quad s_{pq} = \langle p_s, q_s \rangle,
$$
$$
D = a^2 b^2 - 2Kabs_{pq} + K^2 n_p n_q, \quad N = a^2 n_q + 2abs_{pq} + b^2 n_p.
$$

## C.3 LORENTZ NON-LINEAR ACTIVATION

The Lorentz ELU activation applies the activation function to the space components and concatenates them with the time component:

$$
C_{\text{activated}} = \begin{bmatrix} \sqrt{\|\text{ELU}(p_s)\|^2 - 1/K} \\ \text{ELU}(p_s) \end{bmatrix}.
\tag{21}
$$

## C.4 LORENTZ CONCATENATION

Given a set of hyperbolic points $\{p_i \in \mathbb{L}_K^n\}_{i=1}^N$, the Lorentz direct concatenation is defined as:

$$
\boldsymbol{y} = \text{HCat}(\{p_i\}_{i=1}^N) = \begin{bmatrix} \sqrt{\sum_{i=1}^N p_{it}^2 + \dfrac{N-1}{K}}, p_{1s}^T, \ldots, p_{Ns}^T \end{bmatrix}^T,
\tag{22}
$$

where $\boldsymbol{y} \in \mathbb{L}_K^{nN} \subset \mathbb{R}^{nN+1}$.

## C.5 LORENTZ FULLY-CONNECTED LAYER

Let $p \in \mathbb{L}_K^n$ denote the input vector and $\mathbf{W} \in \mathbb{R}^{m \times n+1}$, $v \in \mathbb{R}^{n+1}$ the weight parameters, the Lorentz fully-connected layer (LFC) is defined as:

$$
y = \text{LFC}(p) = \begin{bmatrix} \sqrt{\|\psi(\mathbf{W}p + \mathbf{b})\|^2 - 1/K} \\ \psi(\mathbf{W}p + \mathbf{b}) \end{bmatrix}
\tag{23}
$$

$$
\phi(\mathbf{W}p, v) = \lambda \sigma(\mathbf{v}^T p + b') \frac{\mathbf{W}\psi(p) + \mathbf{b}}{\|\mathbf{W}\psi(p) + \mathbf{b}\|}
\tag{24}
$$

where $\lambda > 0$ denotes a trainable scaling factor, $\mathbf{b} \in \mathbb{R}^n$ is the bias vector, and $\psi$ and $\sigma$ represent the activation and sigmoid functions, respectively.

## C.6 LORENTZ CONVOLUTIONAL LAYER

Given a hyperbolic input feature map $p = \{\mathbf{p}_{h,w} \in \mathbb{L}_K^n\}_{h,w=1}^{H,W}$ as an ordered set of $n$-dimensional hyperbolic feature vectors, the features within the receptive field of the kernel $\mathbf{K} \in \mathbb{R}^{m \times n \times \bar{H} \times \bar{W}}$ are $\{\mathbf{p}_{h'+\delta_{\tilde{h}}, \, w'+\delta_{\tilde{w}}} \in \mathbb{L}_K^n\}_{\tilde{h},\tilde{w}=1}^{\tilde{H},\tilde{W}}$, where $(h', w')$ denotes the starting position and $\delta$ is the stride parameter. The Lorentz convolutional layer is defined as $\text{LFC}\big(\text{HCAT}(\{\mathbf{p}_{h'+\delta_{\tilde{h}}, \, w'+\delta_{\tilde{w}}} \in \mathbb{L}_K^n\}_{\tilde{h},\tilde{w}=1}^{\tilde{H},\tilde{W}})\big)$, where HCAT and LFC, denote hyperbolic concatenation and a Lorentz fully-connected layer performing the affine transformation and parameterizing the kernel and bias.

### C.7 LORENTZ MULTINOMIAL LOGISTIC REGRESSION

Similar to the Euclidean MLR, the Lorentz MLR performs classification by measuring distances to decision hyperplanes. The output logit for class $c$ is computed from the hyperbolic distance between $p$ and its corresponding hyperplane. For a hyperbolic input point $p \in \mathbb{L}_K^n$ and $C$ possible classes, each class $c \in \{1, \dots, C\}$ is associated with a decision hyperplane parameterized by $a_c \in \mathbb{R}$ and $z_c \in \mathbb{R}^n$.

$$v_{z_c, a_c}(p) = \frac{1}{\sqrt{-K}} \text{sign}(\alpha_c) \beta_c \left| \sinh^{-1}\left( \sqrt{-K} \frac{\alpha_c}{\beta_c} \right) \right|, \tag{25}$$

where

$$\alpha_c = \cosh(\sqrt{-K}a_c)\langle z_c, p_s \rangle - \sinh(\sqrt{-K}a_c)\|z_c\|p_t,$$

$$\beta_c = \sqrt{\|\cosh(\sqrt{-K}a_c)z_c\|^2 - (\sinh(\sqrt{-K}a_c)\|z_c\|)^2}.$$

### C.8 $\delta$-HYPERBOLICITY

Khrulkov et al. (2020) introduced $\delta$-hyperbolicity as a measure of the degree of tree-like structure inherent in embeddings. The idea is to find the smallest value of $\delta$ for which the triangle inequality holds via the Gromov product. In this formulation, the definition of a hyperbolic space in terms of the Gromov product can be seen as the metric relations between any four points are the same as they would be in a tree, up to an additive constant $\delta$. Formally, given the Lorentz model $\mathbb{L}_K^n$ with distance $d_{\mathcal{L}}$, the Gromov product of $z, q \in \mathbb{L}_K^n$ with respect to $p \in \mathbb{L}_K^n$ as:

$$(p, q)_z = \frac{1}{2}\big(d_{\mathcal{L}}(p, z) + d_{\mathcal{L}}(q, z) - d_{\mathcal{L}}(p, q)\big). \tag{26}$$

The Lorentz model is said to be $\delta$-hyperbolic for some $\delta \geq 0$ if it satisfies the four-point condition, which states that for any $p, q, z, w \in \mathbb{L}_K^n$:

$$(p, q)_w \geq \min\{(p, z)_w, (q, z)_w\} - \delta. \tag{27}$$

The metric relations between any four points are the same as they would be in a tree, up to the additive constant $\delta$. The lower $\delta \geq 0$ is, the higher the hyperbolicity of the embedding.

# D HEEGNET DETAILS

## D.1 ALGORITHM

---

**Algorithm 1** Hyperbolic domain-specific momentum batch normalization (HDSMBN)

---

**Input:**
batch $\mathcal{B}_k = \{p_i \in \mathbb{L}_K^n, d(i) \in \mathcal{D}_{\mathcal{B}_k}\}_{i=1}^M$ at training step $k$, $d(i)$ indicates the associated domain d
domain-specific running mean $\tilde{\mu}_{k-1}^d (\tilde{\mu}_0^d = \bar{0})$ and variance$\tilde{\nu}^2{}_{k-1}^d (\tilde{\nu}^2{}_0^d = 1)$ for training
domain-specific running mean $\hat{\mu}_{k-1}^d (\hat{\mu}_0^d = \bar{0})$ and variance$\hat{\nu}^2{}_{k-1}^d (\hat{\nu}^2{}_0^d = 1)$ for testing
momentum for training and testing $\eta_{train(k)}, \eta_{test} \in [0, 1]$, learnable parameter $\nu_\phi^2$

**Output:** normalized batch $\{\tilde{p}_i\} = \text{HBN}^{d(i)}(p_i)$
    **if** training **then then**
        Compute domain-specific batch mean $\mu_k^d$ and variance $\nu^2{}_k^d$         ▷ using Eq. (2)
        $\tilde{\mu}_k^d = \text{wFM}_{\eta_{train(k)}}(\tilde{\mu}_{k-1}^d, \mu_k^d)$         ▷ update running mean using Eq. (2)
        $\tilde{\nu}^2{}_k^d = (1 - \eta_{train(k)})\, \tilde{\nu}^2{}_{k-1}^d + \eta_{train(k)}\, \nu^2{}_k^d$
        $\hat{\mu}_k^d = \text{wFM}_{\eta_{test}}(\hat{\mu}_{k-1}^d, \mu_k^d)$         ▷ update running mean using Eq. (2)
        $\hat{\nu}^2{}_k^d = (1 - \eta_{test})\, \hat{\nu}^2{}_{k-1}^d + \eta_{test}\, \nu^2{}_k^d$
    **end if**
    $(\mu_k^d, \nu^2{}_k^d) \leftarrow (\tilde{\mu}_k^d, \tilde{\nu}^2{}_k^d)$ **if training else** $(\hat{\mu}_k^d, \hat{\nu}^2{}_k^d)$
    $\tilde{p}_i = \frac{\nu_\phi^2}{\sqrt{\nu^2{}_k^d + \epsilon}} \big( \ominus\, \mu_k^d \oplus p_i \big)$         ▷ use Eq. (7) to recentering and rescale each domain

---

---

**Algorithm 2** Horospherical Hyperbolic Sliced-Wasserstein loss (HHSW)

---

**Input:**
batch $\mathcal{B} = \{\tilde{p}_i \in \mathbb{L}_K^n, d(i) \in \mathcal{D}_{\mathcal{B}}\}_{i=1}^M$, $d(i)$ indicates the associated domain $d$
number of slices $S = 1000$, exponent $p = 2$

**Output:** scalar loss $\mathcal{L}_{\text{swd}}$
    Initialize $\mathcal{L}_{\text{swd}} \leftarrow 0$
    **for** each domain $d \in \mathcal{D}_{\mathcal{B}_k}$ **do**
        Extract domain-specific samples $\mathcal{P}^d = \{p_i \mid d(i) = d\}$
        Sample Gaussian noise $Z^d \sim \mathcal{N}(0, I)$ with shape shape($\mathcal{P}^d$)
        Normalize: $Z^d \leftarrow \frac{Z^d}{\|Z^d\|_2 + \epsilon}$
        Map to hyperbolic manifold: $Q^d \leftarrow \exp_0^K(Z^d)$
        Compute domain loss: $\ell^d \leftarrow \text{HHSW}_p^p(\mathcal{P}^d, Q^d)$         ▷ Eq. (6)
        $\mathcal{L}_{\text{swd}} \leftarrow \mathcal{L}_{\text{swd}} + \ell^d$
    **end for**
    **return** $\mathcal{L}_{\text{swd}}$

---

## D.2 MODEL ARCHITECTURE

Table 6: HEEGNet architecture details. $P$: electrodes; $T$: temporal samples; $C$: classes.

| Layer | Output (dim) | Parameter (dim) | Operation | Space |
|---|---|---|---|---|
| *Input:* $1 \times P \times T$ | | | | |
| TempConv | $8 \times P \times T$ | $8 \times 1 \times 1 \times 64$ | convolution | Euclidean |
| BN | $8 \times P \times T$ | $8$ | batch norm | Euclidean |
| SpatConv | $16 \times 1 \times T$ | $16 \times 8 \times P \times 1$ | depthwise conv | Euclidean |
| BN | $16 \times 1 \times T$ | $16$ | batch norm | Euclidean |
| Activation | $16 \times 1 \times T$ | – | ELU | Euclidean |
| AvgPool | $16 \times 1 \times \lfloor T/4 \rfloor$ | – | pooling | Euclidean |
| Dropout | $16 \times 1 \times \lfloor T/4 \rfloor$ | – | dropout=0.25 | Euclidean |
| DepthConv | $16 \times 1 \times \lfloor T/4 \rfloor$ | $16 \times 1 \times 1 \times 16$ | depthwise conv | Euclidean |
| ProjX | $17 \times \lfloor T/4 \rfloor$ | – | projection | Euclidean |
| PointConv | $17 \times \lfloor T/4 \rfloor$ | – | pointwise conv | Hyperbolic |
| DSMDBN$_{(1)}$ | $17 \times \lfloor T/4 \rfloor$ | – | DSMDBN | Hyperbolic |
| Activation | $17 \times \lfloor T/4 \rfloor$ | – | ELU | Hyperbolic |
| AvgPool | $17 \times \lfloor T/32 \rfloor$ | – | pooling | Hyperbolic |
| Flatten | $16 \cdot \lfloor T/32 \rfloor + 1$ | – | flatten | Hyperbolic |
| MLR | $C$ | $(16 \cdot \lfloor T/32 \rfloor + 1) \times C$ | - | Hyperbolic |

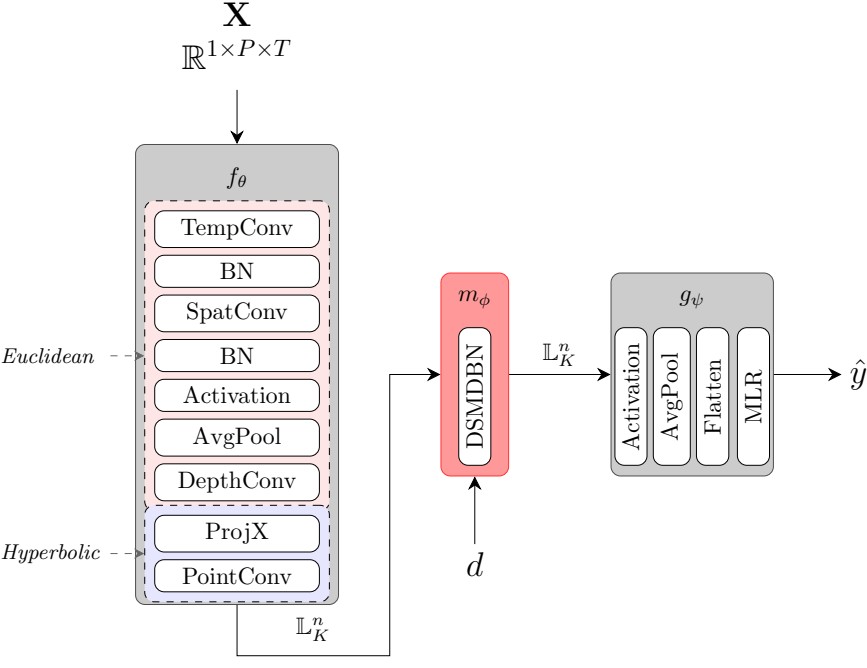

Figure 3: HEEGNet architecture.

## D.3 SOFTWARE AND HARDWARE

We used publicly available Python code for baseline methods and implemented custom methods using the packages torch (Paszke et al., 2019), scikit-learn (Pedregosa et al., 2011), geoopt (Kochurov et al., 2020). We conducted all experiments on standard computation PCs with 32-core CPUs, 128 GB of RAM, and a single GPU.

## D.4 COMPUTATIONAL COST

Hyperbolic neural networks remain in their early development stage and introduce additional computational cost, because of the need for exponential and logarithmic mappings. To evaluate the practical computational cost and the efficiency of our implementations, we compared the per-epoch runtimes of four models: the original EEGNet, HEEGNet without SFUDA, HEEGNet with moments alignment ($DSMDBN_1$), and HEEGNet with DSMDBN.

Table 7: **Hyperbolic computational costs**. Comparisons of the runtime (seconds) per epoch.

| Model | SFUDA | Cost (seconds/epoch) |
|---|---|---|
| EEGNet | w/o | 0.420 |
| HEEGNet | w/o | 1.157 |
| HEEGNet | $DSMDBN_1$ | 1.320 |
| HEEGNet | DSMDBN | 1.451 |

# E  FULL EXPERIMENTS

## E.1  DATASET DETAILS

**Nakanishi2015** is an SSVEP dataset including EEG from 9 subjects recorded with 8 channels at 256 Hz. Each subject performed a 12-class joint frequency-phase modulation paradigm with 15 trials per class, each trial lasting 4.15 seconds. The dataset consists of a single session per subject and was originally designed to evaluate online BCI performance. This benchmark is widely used for 12-class SSVEP decoding.

**Wang2016** is an SSVEP dataset including EEG from 34 subjects recorded with 64 channels at 250 Hz. Each subject performed a 40-class visual stimulation paradigm, where 40 flickering targets were presented. The experiment comprised 6 blocks per subject, with 40 trials per block and 6 trials per class in total. Each trial lasted 6 seconds, and subjects were instructed to gaze at the cued target while avoiding blinks during stimulation. This dataset provides a large-scale benchmark for 40-class SSVEP decoding.

**CBVEP** is a c-VEP and burst-VEP dataset including EEG from 12 subjects recorded with 32 channels at 500 Hz. Each subject performed a 4-class visual stimulation paradigm, with 15 trials per class and trial duration of 2.2 seconds. EEG was recorded using a BrainProduct LiveAmp 32 system with electrodes placed according to the 10–20 system, referenced to FCz and grounded at FPz. Participants focused on cued targets during each stimulation phase, and post-experiment subjective ratings of visual comfort, tiredness, and intrusiveness were also collected. This dataset provides a benchmark for 4-class c-VEP decoding.

**SEED** is an emotion recognition dataset including EEG and eye movement data from 15 subjects, recorded with a 62-channel system. Each subject performed a 3-class emotion elicitation task, where they watched 15 different film clips (approximately 4 minutes each) designed to evoke positive, neutral, or negative emotions. The experiment consisted of 15 trials, with each trial including a 5 s hint before the clip, 45 s for self-assessment, and a 15 s rest period afterward.

**Faced** is a fine-grained emotion recognition dataset including EEG from 123 subjects recorded with 32 channels at 250 Hz. Each subject performed a 9-class emotion elicitation task, where they watched 28 video clips selected to evoke a range of emotions including amusement, inspiration, joy, tenderness, anger, fear, disgust, sadness, and neutral states. This dataset provides a large-scale, fine-grained, and balanced benchmark for 9-class emotion recognition from EEG signals.

**Boran** is an intracranial EEG (iEEG) dataset including recordings from 9 patients with drug-resistant focal epilepsy. The dataset contains simultaneous recordings from stereotactically implanted depth electrodes in the medial temporal lobe and from scalp EEG electrodes placed according to the 10–20 system. Macroelectrode iEEG was recorded at 4 kHz and microelectrode iEEG at 32 kHz, while scalp EEG was recorded at 256 Hz. Recordings were performed using an ATLAS system for iEEG and a NicoletOne system for scalp EEG. The dataset provides high-resolution iEEG and single-neuron data from the human medial temporal lobe for studying epilepsy.

**BNCI2015001** (Faller et al., 2012) is a motor imagery dataset including EEG recordings from 12 subjects with 13 channels at 512 Hz. Each subject performed sustained right hand versus both feet motor imagery across 200 trials per class, resulting in a total of 14,400 trials. The experiment comprised 3 sessions, each with a single run of 5-second trials. EEG was recorded from Laplacian derivations centered on C3, Cz, and C4 according to the international 10–20 system.

**BNCI2014001** (Tangermann et al., 2012) is a motor imagery dataset containing EEG from 9 subjects recorded with 22 channels at 250 Hz. The paradigm involves four motor imagery tasks: left hand, right hand, both feet, and tongue, with 144 trials per class per subject. Each subject completed two sessions on different days, each comprising 6 runs of 48 4-second trials.

E.2 VISUALLY EVOKED POTENTIALS

Table 8: **VEP per dataset results.** Average of test-set scores (balanced accuracy (%); higher is better; mean ± std).

| Model | SFUDA | Dataset | | | |
|---|---|---|---|---|---|
| | | CBVEP100 | CBVEP40 | Nakanishi | Wang |
| ATCNet | w/o | 25.4 ± 1.8 | 25.2 ± 3.7 | 56.3 ± 20.6 | 32.8 ± 13.1 |
| | EA | 74.4 ± 12.1 | 74.6 ± 9.0 | 54.7 ± 18.9 | 35.8 ± 10.8 |
| | STMA | 25.6 ± 1.5 | 24.5 ± 1.0 | 58.9 ± 18.7 | 44.3 ± 11.1 |
| EEGConformer | w/o | 25.0 ± 0.0 | 25.1 ± 0.2 | 29.4 ± 10.2 | 31.9 ± 11.0 |
| | EA | 75.8 ± 10.3 | 73.1 ± 9.6 | 29.4 ± 9.0 | 30.5 ± 8.5 |
| | STMA | 25.0 ± 0.0 | 24.8 ± 0.5 | 24.3 ± 7.4 | 30.7 ± 8.2 |
| EEGNet | w/o | 24.9 ± 0.9 | 25.7 ± 4.9 | 57.2 ± 19.8 | 37.4 ± 12.9 |
| | EA | 79.5 ± 11.2 | 77.4 ± 8.2 | 56.0 ± 16.5 | 39.7 ± 9.5 |
| | STMA | 24.6 ± 2.9 | 24.1 ± 3.3 | 40.0 ± 10.7 | 32.0 ± 8.4 |
| FBCNet | w/o | 29.4 ± 3.9 | 30.4 ± 3.1 | 28.3 ± 7.5 | 16.4 ± 4.5 |
| | EA | 75.8 ± 10.4 | 74.3 ± 10.1 | 27.7 ± 7.4 | 16.2 ± 3.0 |
| | STMA | 25.0 ± 0.5 | 24.2 ± 2.4 | 27.9 ± 7.3 | 17.8 ± 3.4 |
| ShallowNet | w/o | 76.2 ± 7.2 | 74.3 ± 8.4 | 29.4 ± 7.7 | 18.4 ± 6.7 |
| | EA | 66.7 ± 14.8 | 64.9 ± 13.7 | 27.6 ± 7.2 | 16.7 ± 4.4 |
| | STMA | 54.7 ± 8.3 | 52.5 ± 6.4 | 28.6 ± 8.3 | 19.8 ± 4.3 |
| TSLANet | w/o | 30.4 ± 12.5 | 41.8 ± 19.4 | 16.1 ± 4.3 | 18.4 ± 6.2 |
| | EA | 73.5 ± 9.6 | 73.5 ± 9.1 | 14.0 ± 1.7 | 18.0 ± 5.5 |
| | STMA | 25.9 ± 1.7 | 26.3 ± 2.4 | 15.4 ± 3.3 | 17.8 ± 4.0 |
| DDGCNN | w/o | 24.4 ± 1.8 | 25.5 ± 2.2 | 26.2 ± 12.9 | 30.3 ± 10.7 |
| | EA | 72.2 ± 11.1 | 70.8 ± 11.4 | 54.7 ± 21.2 | 46.1 ± 12.2 |
| EEGInception | w/o | 28.9 ± 5.5 | 34.2 ± 20.9 | 60.7 ± 22.0 | 38.7 ± 12.8 |
| | EA | 79.9 ± 8.8 | 75.5 ± 11.1 | 61.5 ± 19.0 | 41.8 ± 9.6 |
| SSVEPNet | w/o | 26.0 ± 3.8 | 28.1 ± 4.4 | 70.8 ± 20.5 | 55.1 ± 16.0 |
| | EA | 72.4 ± 13.3 | 68.5 ± 10.9 | 68.9 ± 20.2 | 53.9 ± 13.0 |
| SSVEPFormer | w/o | 24.0 ± 3.9 | 26.7 ± 6.4 | 77.2 ± 21.0 | 72.3 ± 15.6 |
| | EA | 72.2 ± 8.4 | 72.1 ± 7.0 | 77.0 ± 19.8 | 72.6 ± 12.6 |
| LaBraM | w/o | 63.9 ± 26.0 | 61.8 ± 23.5 | 69.9 ± 25.5 | 72.2 ± 13.6 |
| CBraMod | w/o | 62.0 ± 13.9 | 59.3 ± 10.5 | 68.6 ± 14.3 | 70.7 ± 21.9 |
| HEEGNet | DSMDBN | **95.8 ± 6.2** | **92.6 ± 22.1** | 79.8 ± 20.8 | 69.3 ± 15.5 |
| | DSMDBN+EA | 83.8 ± 9.5 | 79.0 ± 10.2 | **81.9 ± 18.7** | **73.5 ± 11.4** |

## E.3 EMOTION RECOGNITION

Table 9: **Emotion recognition per dataset results.** Average of test-set scores (balanced accuracy (%); higher is better; mean ± std).

| Model | SFUDA | Dataset | |
|---|---|---|---|
| | | Faced | Seed |
| ATCNet | w/o | $11.2 \pm 0.9$ | $34.8 \pm 9.8$ |
| | EA | $57.5 \pm 15.0$ | $44.6 \pm 13.1$ |
| | STMA | $59.3 \pm 15.5$ | $48.4 \pm 15.7$ |
| EEGConformer | w/o | $85.4 \pm 14.3$ | $77.5 \pm 21.7$ |
| | EA | $89.0 \pm 11.1$ | $65.9 \pm 18.1$ |
| | STMA | $81.5 \pm 13.9$ | $69.2 \pm 17.7$ |
| EEGNet | w/o | $24.8 \pm 13.7$ | $74.5 \pm 19.8$ |
| | EA | $83.9 \pm 14.5$ | $46.2 \pm 15.4$ |
| | STMA | $78.2 \pm 16.6$ | $50.1 \pm 15.2$ |
| FBCNet | w/o | $19.0 \pm 8.4$ | $54.5 \pm 17.4$ |
| | EA | $42.1 \pm 12.7$ | $76.3 \pm 17.4$ |
| | STMA | $23.1 \pm 8.9$ | $60.0 \pm 20.3$ |
| ShallowNet | w/o | $43.1 \pm 16.6$ | $76.2 \pm 17.6$ |
| | EA | $80.3 \pm 16.7$ | $75.6 \pm 15.2$ |
| | STMA | $64.6 \pm 17.2$ | $63.3 \pm 15.5$ |
| TSLANet | w/o | $33.3 \pm 11.8$ | $41.8 \pm 13.8$ |
| | EA | $37.2 \pm 12.4$ | $65.6 \pm 14.4$ |
| | STMA | $46.2 \pm 13.1$ | $58.7 \pm 16.4$ |
| EMT | w/o | $32.9 \pm 13.4$ | $48.4 \pm 14.5$ |
| | EA | $38.6 \pm 11.4$ | $42.8 \pm 16.1$ |
| TSception | w/o | $14.3 \pm 4.6$ | $42.2 \pm 15.4$ |
| | EA | $84.3 \pm 13.3$ | $55.0 \pm 18.0$ |
| CBraMod | w/o | $48.3 \pm 7.3$ | $70.9 \pm 18.7$ |
| LaBraM | w/o | $52.4 \pm 11.6$ | $73.5 \pm 17.6$ |
| HEEGNet | DSMDBN | $84.1 \pm 14.7$ | $\textbf{79.4} \pm \textbf{20.8}$ |
| | DSMDBN+EA | $\textbf{89.7} \pm \textbf{11.2}$ | $78.7 \pm 12.4$ |

### E.4 INTRACRANIAL EEG RESULTS

Table 10: **Intracranial EEG result.** Average of test-set scores (balanced accuracy (%); higher is better; mean ± std).

| Model | SFUDA | Intracranial |
|---|---|---|
| EEGNet | w/o | $55.4 \pm 9.1$ |
| | EA | $59.1 \pm 11.2$ |
| | STMA | $51.2 \pm 5.9$ |
| EEGConformer | w/o | $55.5 \pm 7.0$ |
| | EA | $61.7 \pm 13.1$ |
| | STMA | $53.1 \pm 5.2$ |
| ATCNet | w/o | $57.5 \pm 9.0$ |
| | EA | $57.1 \pm 9.0$ |
| | STMA | $54.2 \pm 5.7$ |
| TSLANet | w/o | $55.1 \pm 9.3$ |
| | EA | $54.9 \pm 10.4$ |
| | STMA | $54.1 \pm 6.2$ |
| SchirrmeisterNet | w/o | $56.9 \pm 7.7$ |
| | EA | $65.0 \pm 10.2$ |
| | STMA | $53.9 \pm 5.5$ |
| FBCNet | w/o | $61.1 \pm 10.6$ |
| | EA | $61.3 \pm 11.0$ |
| | STMA | $53.0 \pm 6.8$ |
| LaBraM | w/o | $44.4 \pm 49.7$ |
| CBraMod | w/o | $53.6 \pm 6.2$ |
| HEEGNet (proposed) | DSMDBN | $58.4 \pm 9.6$ |
| | DSMDBN+EA | $\mathbf{66.7 \pm 12.5}$ |

## F   FULLY HYPERBOLIC NEURAL NETWORK EXPERIMENT

To investigate the necessity of the hybrid design in HEEGNet, we conducted experiments by replacing Euclidean convolutional operations in EEGNet (Lawhern et al., 2018) with hyperbolic variants as follows:

- **None (EEGNet)** : Original EEGNet
- **HMLR**: Replaced the final MLR layer with hyperbolic MLR
- **HMLR + Block2**: Additionally replaced Block2 in EEGNet (the depht-wise convolution and the point-wise convolution)
- **Full Hyperbolic**: Fully hyperbolic models

As shown in Tab. 11, the performance on the 4 representative datasets demonstrates a degradation pattern as more Euclidean operations are replaced with hyperbolic variants. These result suggests that the Euclidean backbone is essential for effective downstream classification, supporting the rationale behind our hybrid architecture, which combines Euclidean encoders with hyperbolic neural networks.

Table 11: **Fully hyperbolic neural network experiments**. The averages of test-set scores are shown above using balanced accuracy (the score and the standard deviation are shown for each dataset). The number of domains in each dataset is indicated by $n$.

| Dataset Model | Visual | | Emotion | Intracranial |
|---|---|---|---|---|
| | Nakanishi $(n = 9)$ | Wang $(n = 34)$ | Seed $(n = 45)$ | Boran $(n = 37)$ |
| None (EEGNet) | $57.2 \pm 19.8$ | $37.4 \pm 12.9$ | $74.5 \pm 19.8$ | $55.4 \pm 9.1$ |
| HMLR | $60.8 \pm 20.7$ | $39.2 \pm 13.8$ | $75.1 \pm 20.0$ | $57.4 \pm 8.7$ |
| HMLR + Block2 | $53.4 \pm 17.5$ | $32.5 \pm 11.9$ | $76.9 \pm 22.1$ | $55.5 \pm 7.3$ |
| Fully Hyperbolic | $27.3 \pm 12.0$ | $22.5 \pm 8.8$ | $70.7 \pm 21.6$ | $52.7 \pm 8.9$ |

