# OpenReview forum: "HEEGNet: Hyperbolic Embeddings for EEG"
_ICLR.cc/2026/Conference — ICLR 2026 Poster_

### Official Review · Reviewer_HQzc · 2025-10-30

**Soundness:** 2
**Presentation:** 3
**Contribution:** 3
**Rating:** 6
**Confidence:** 4

**Summary:**

This work focuses on the hyperbolic representation of the EEG signal to enhance classification tasks. Based on the demonstration of EEG showing hyperbolicity to improve generalization, the authors propose HEEGNet to capture hierarchical structure in EEG and learn domain-invariant information. HEEGNet achieves superior results in various EEG classification tasks.

**Strengths:**

1. The work leveraged hyperbolic geometry to represent EEG features, which gave technical novelty and a good basis for EEG classification tasks.
2. Extensive evaluations have been performed to demonstrate the usability of the HEEGNet. Various SOTA methods were involved for comparison.
3. The authors gave a clear description of the model and source code for reproduction.

**Weaknesses:**

1. The roadmap on why a hyperbolic manifold is necessary for EEG and further comparison with other methods is necessary to show the improvement of the current method.
2. The overall performance of HEEGNet outperformed other methods. However, it's not clear which part of the model contributes more to the final results.
3. Apart from the classification results, it's not clear if the hyperbolic embeddings could help us find brain patterns precisely. Further analysis would be beneficial.
4. The motor imagery paradigm is an important paradigm to explore the significance of EEG geometry. It would be better to include the motor imagery comparison in the manuscript instead of the Appendix.
5. Several datasets from different paradigms were included for comparison. It would be better to clarify the validation manner to help understand the use of the methods. Is HEEGNet also useful in the generalization cases, such as cross-subject classification?

**Questions:**

1. Is the model only suitable for the EEGNet used in the paper, or also works for other methods like spatial-temporal convolution and some self-attention?
2. How much computational cost is taken by the hyperbolic calculation?
3. Is there a Discussion section for the work?

---

> ### Author Response · Authors · 2025-11-19
> **Response [1/3] to Official Review by Reviewer HQzc - weaknesses**
>
> Thank you very much for your effort to assess our submission and the provided feedback.
> For your convenience, we include copies of the revised manuscript [manuscript_revised.pdf](https://anonymous.4open.science/r/HEEGNet-F655/revised_version.pdf) and the submitted manuscript [manuscript_submitted.pdf](https://anonymous.4open.science/r/HEEGNet-F655/submitted_version.pdf).
>
> Please find our detailed responses to the weaknesses and questions below.
>
> > **The roadmap on why a hyperbolic manifold is necessary for EEG**
>
> Thank you for sharing your concern.
>
> Our motivation is grounded in neuroscientific evidence showing that EEG signals can reflect hierarchical cognitive processing [1][2][3].
> These findings provide a well-established basis for considering hierarchical structure in EEG representations.
>
> For empirical justification, we employ $\delta$-hyperbolicity [4][5], a well-established metric to determine hierarchical and tree-like structures (See an explanation in Appendix C.8).
> In our experiments (Table 1), we compute $\delta$-hyperbolicity of raw EEG data, EEGNet-generated embeddings from intermediate layers (after the first two convolutional layers), and the classification space embeddings.
> We observed that both raw EEG data and EEGNet-generated embeddings exhibit hierarchical structures.
>
> Prior work [6][7] in hyperbolic manifolds further shows that hyperbolic geometry is well-suited for any data whose latent structure is tree-like or hierarchical, regardless of modality.
> Our results confirm that hyperbolic embeddings improve cross-domain generalization (Table 2)and enhance class separability (Figure 2), providing strong empirical support for using hyperbolic geometry in EEG.
>
>
> *References*:
>
> [1] Elliot Collins, Amanda K Robinson, and Marlene Behrmann. Distinct neural processes for the perception of familiar versus unfamiliar faces along the visual hierarchy revealed by eeg. NeuroImage, 181:120-131, 2018. doi: 10.1016/j.neuroimage.2018.06.080.
>
> [2] Sai Sun, Hongbo Yu, Rongjun Yu, and Shuo Wang. Functional connectivity between the amygdala and prefrontal cortex underlies processing of emotion ambiguity. Translational psychiatry, 13(1): 334, 2023. doi: 10.1038/541398-023-02625-W.
>
> [3] William Turner, Tessel Blom, and Hinze Hogendoorn. Visual information is predictively encoded in occipital alpha/low-beta oscillations. Journal of Neuroscience, 43(30):5537-5545, 2023. doi: 10.1523/JNEUROSCI.0135-23.2023.
>
> [4] Valentin Khrulkov, Leyla Mirvakhabova, Evgeniya Ustinova, Ivan Oseledets, and Victor Lempitsky. Hyperbolic image embeddings. In Proceedings of the IEEE/CVF conference on computer vision and pattern recognition, pp. 6418-6428, 2020.
>
> [5] Ahmad Bdeir, Kristian Schwethelm, and Niels Landwehr. Fully hyperbolic convolutional neural networks for computer vision. In ICLR, 2024
>
> [6] Yang, Menglin, et al. "Hyperbolic representation learning: Revisiting and advancing." International Conference on Machine Learning. PMLR, 2023.
>
> [7] Peng, Wei, et al. "Hyperbolic deep neural networks: A survey." IEEE Transactions on pattern analysis and machine intelligence 44.12 (2021): 10023-10044.

---

> ### Author Response · Authors · 2025-11-19
> **Response [2/3] to Official Review by Reviewer HQzc - weaknesses**
>
> > **further comparison with other methods is necessary to show the improvement of the current method.**
>
> Thank you for your suggestions.
> We now include manifold-based decoding approaches for motor imagery in Table 5.
> These include Grassmann manifold methods: GDLNet[1] and GyroAtt-Gr[2]; Symmetric Positive Definite manifold methods: TSMNet[3], MAtt[4], and CSPNet[5]; as well as Symmetric Positive Semi-Definite manifold method: GyroAtt-SPSD[2].
> Requested by reviewer VB3w, we also include foundation model LaBraM[6] and CBraMod[7] as baselines.
> Our proposed HEEGNet achieved competitive performance and state-of-the-art results in most datasets.
>
> *References*:
>
> [1] Rui Wang, Chen Hu, Ziheng Chen, Xiao-Jun Wu, and Xiaoning Song. A grassmannian manifold self-attention network for signal classification. In Proceedings of the Thirty-Third International Joint Conference on Artificial Intelligence, pp. 5099-5107, 2024
>
> [2] Rui Wang, Chen Hu, Xiaoning Song, Xiaojun Wu, Nicu Sebe, and Ziheng Chen. Towards a general attention framework on gyrovector spaces for matrix manifolds. In The Thirty-ninth Annual Conference on Neural Information Processing Systems, 2025
>
> [3] Reinmar Kobler, Jun-ichiro Hirayama, Qibin Zhao, and Motoaki Kawanabe. Spd domain-specific batch normalization to crack interpretable unsupervised domain adaptation in eeg. Advances in Neural Information Processing Systems, 35:6219-6235, 2022.
>
> [4] Yue-Ting Pan, Jing-Lun Chou, and Chun-Shu Wei. Matt: A manifold attention network for eeg decoding. In Advances in Neural Information Processing Systems, volume 35, pp. 31116-31129, 2022
>
> [5] Ce Ju and Cuntai Guan. Tensor-cspnet: A novel geometric deep learning framework for motor imagery classification. IEEE Transactions on Neural Networks and Learning Systems, 34(12): 10955-10969, 2022.
>
> [6] Jiang, Weibang, Liming Zhao, and Bao-liang Lu. "Large Brain Model for Learning Generic Representations with Tremendous EEG Data in BCI." The Twelfth International Conference on Learning Representations.
>
> [7] Wang, Jiquan, et al. "CBraMod: A Criss-Cross Brain Foundation Model for EEG Decoding." The Thirteenth International Conference on Learning Representations.

---

> ### Author Response · Authors · 2025-11-19
> **Response [3/3] to Official Review by Reviewer HQzc - weaknesses**
>
> > **The overall performance of HEEGNet outperformed other methods. However, it's not clear which part of the model contributes more to the final results.**
>
> Thank you for your comments.
>
> The primary performance gains can be attributed to two key components: hyperbolic embeddings and our novel domain adaptation strategy.
> We conducted an ablation study (Table 2) to evaluate the effectiveness of hyperbolic embeddings.
> The results show that hyperbolic embeddings consistently improve cross-domain generalization. A second ablation study (Table 4) compares the three alignment stages within our domain adaptation strategy. The results demonstrate systematic performance improvements as each stage is added, confirming the complementary benefits of our multi-stage alignment design.
>
> In addition, we visualize the same subject and same session under different training configurations—Euclidean embeddings, hyperbolic multinomial logistic regression, moment alignment, and DSMDBN (proposed).
> This controlled setup allows us to directly compare the relative advantages of hyperbolic geometry and our adaptation modules, and we observe a systematic enhancement of class separability across configurations.
>
> > **Apart from the classification results, it's not clear if the hyperbolic embeddings could help us find brain patterns precisely. Further analysis would be beneficial.**
>
> Thank you for your comments.
> We demonstrated that hyperbolic embeddings capture hierarchical EEG structure through multiple lines of evidence:
> - We confirms that EEG data exhibits hierarchical structures (Table 1).
> - We provide t-SNE visualizations, Figure 2a and Figure 2b, that visually confirm that hyperbolic representations enhance class separability, indicating a more meaningful organization of task-relevant brain states.
> - The consistent improvement across all datasets (Table 2) demonstrates that hyperbolic embeddings better preserve class structure in the latent space than Euclidean ones.
>
> We would like to kindly note that the primary focus of this work is the integration of hyperbolic geometry with our novel domain adaptation strategy to achieve robust cross-domain generalization in EEG decoding.
>
>
> > **The motor imagery paradigm is an important paradigm to explore the significance of EEG geometry. It would be better to include the motor imagery comparison in the manuscript instead of the Appendix.**
>
> Thank you for your suggestions. We now include motor imagery comparison with more manifold learning methods in the main text of the revised manuscript, in section 4.3.
>
>
> > **Several datasets from different paradigms were included for comparison. It would be better to clarify the validation manner to help understand the use of the methods. Is HEEGNet also useful in the generalization cases, such as cross-subject classification?**
>
> Thank you for your questions.
> Actually, HEEGNet leverages the high quality of hyperbolic EEG embedding and our novel domain adaptation strategy to obtain domain-invariant features, which is specifically tailored to address both cross-subject and cross-session classification challenges.
>
> For all datasets, we treat sessions as domains, and use either a leave-one-group-out (source domain number $\leq$10) or a 10-fold leave-groups-out cross-validation scheme to fit and evaluate models.
> For the intracranial EEG dataset, due to the different number of electrodes across subjects, we consider the cross-session adaptation setting.
> We fit and evaluate models independently for each subject, treating the session as the grouping variable.
> For other datasets, we consider the cross-subject adaptation setting and treat the subject as the grouping variable.

---

> ### Author Response · Authors · 2025-11-19
> **Response to Official Review by Reviewer HQzc - questions**
>
> > **Is the model only suitable for the EEGNet used in the paper, or also works for other methods like spatial-temporal convolution and some self-attention?**
>
> Interesting questions.
> Our proposed domain adaptation strategy is technically encoder-agnostic.
> Because the method only requires an encoder to produce a feature map that can then be mapped into the hyperbolic space, it can be combined not only with EEGNet but also with other Euclidean encoders。
> We quickly conducted a similar experiment to replace the multinomial logistic regression in EEGConformer and replace it with a hyperbolic variant, and we observed a 3% performance gain on the iEEG dataset.
>
> > **How much computational cost is taken by the hyperbolic calculation?**
>
> Thank you for your question.
> Indeed, hyperbolic operations like exponential and logarithmic mappings introduce additional computational cost.
> For practical computational cost evaluation, we compared the per-epoch runtimes of four models: the original EEGNet 0.420s, HEEGNet without SFUDA 1.157s, HEEGNet with moments alignment (DSMDBN1) 1.320s, and HEEGNet with DSMDBN 1.451s.
> We now include computational analysis in the revised manuscript Appendix D.4.
>
>
> > **Is there a Discussion section for the work?**
>
> Thank you for pointing this out.
> We now include a discussion on our contribution to the hybrid design, novel domain adaptation strategy, and feature directions.

---

### Official Review · Reviewer_Qng1 · 2025-10-30

**Soundness:** 3
**Presentation:** 3
**Contribution:** 3
**Rating:** 4
**Confidence:** 3

**Summary:**

Overall this paper is working on a promising research direction of EEG-based BCIs/recognitions.
However the paper needs to be significantly improved to better clarify both its contributions and performance comparisons.

**Strengths:**

1. The research direction of using hyperbolic embeddings for EEG-based BCIs/recognitions is emerging and promising.
2. The proposed method is well explained.
3. The performance improvements compared with EEGNet are significant.

**Weaknesses:**

1. The novelty is not well clarified; and the idea of exploring hyperbolic space /embeddings for EEG-based recognition is not new. The authors didn't clearly explain the difference.
2. My main concern is about the performance comparison part.
-- The proposed HEEGNet was not compared with other hyperbolic embeddings based methods, e.g., Jing Chang 2025 in the ref list and related refs/citations of that paper.
-- For different tasks/datasets, it doesn't seem that the proposed methods were compared with SOTAs. Why only baselines were compared? The authors should compare their results with SOTA performances of the datasets, e.g., Seed, Faced.
E.g., the results from comparison methods for VEP in table 3 are far away from SOTA performances reported in the literature. The gap between the proposed method and other methods is too good to believe. More clarification is needed.
-- It is also not clear whether the methods are tested in the cross-dataset setting. E.g, for the Emotion recognition task, are the Seed and Faced datasets combined or tested separately?
3. Minor comment: Abstract is a bit misleading. Reviewers would think this is the first work on exploring hyperbolic embeddings for EEG.

**Questions:**

1. The authors should carefully clarify their novelty.
2. The performance comparison part need to be significantly improved and justified.

---

> ### Author Response · Authors · 2025-11-19
> **Response to Official Review by Reviewer Qng1 - weaknesses**
>
> We appreciate the reviewer's effort to provide feedback and suggestions.
> For your convenience, we include copies of the revised manuscript [manuscript_revised.pdf](https://anonymous.4open.science/r/HEEGNet-F655/revised_version.pdf) and the submitted manuscript [manuscript_submitted.pdf](https://anonymous.4open.science/r/HEEGNet-F655/submitted_version.pdf).
>
> Please find our detailed responses to the weaknesses and questions below.
>
> > **The novelty is not well clarified; and the idea of exploring hyperbolic space /embeddings for EEG-based recognition is not new. The authors didn't clearly explain the difference.**
>
> Thank you for sharing your concern.
> Prior work (Chang2025+, IEEE TAC) applied contrastive pretraining in hyperbolic space for cross-subject emotion recognition. However, their approach requires target-domain label information during pretraining.
> In contrast, our method addresses the more challenging problem of unsupervised domain adaptation for general EEG signals without utilizing any target-domain labels at any stage of training.
>
> In our work, we utilize hyperbolic geometry to tackle the unsupervised domain adaptation problem in general EEG signals.
> We first empirically demonstrate that EEG data exhibits hyperbolicity, and that hyperbolic embeddings improve cross-domain generalization and class separability.
> We then propose the hybrid model design of Euclidean encoders and hyperbolic neural networks to balance feature encoding and hierarchical structural representation and obtain high-quality EEG embeddings.
> The quality of embeddings plays a crucial role in domain adaptation; better embeddings generally lead to improved adaptation performance.
> We further propose a domain adaptation strategy, DSMDBN, that combines moments alignment and feature distribution alignment, achieving state-of-the-art results across multiple tasks.
> Through ablation studies, we demonstrate that the effectiveness of hyperbolic embeddings (Table 2) and our novel DSMDBN adaptation strategy (Table 4).
>
> *References*
>
> Jiang Chang, Zhixin Zhang, Yuhua Qian, and Pan Lin. Multi-scale hyperbolic contrastive learning for cross-subject eeg emotion recognition. IEEE Transactions on Affective Computing, 2025. doi:10.1109/TAFFC.2025.3535542.
>
> > **My main concern is about the performance comparison part. -- The proposed HEEGNet was not compared with other hyperbolic embeddings based methods, e.g., Jing Chang 2025 in the ref list and related refs/citations of that paper. --**
>
> Thank you for sharing your concerns.
> Prior work (Chang2025+, IEEE TAC) leverages target domain label information during the pretraining stage. In contrast, our work focuses on unsupervised domain adaptation without utilizing any target domain labels.
> Therefore, the two approaches are based on different problem settings and are not directly comparable.
>
> *References*
>
> Jiang Chang, Zhixin Zhang, Yuhua Qian, and Pan Lin. Multi-scale hyperbolic contrastive learning for cross-subject eeg emotion recognition. IEEE Transactions on Affective Computing, 2025. doi:10.1109/TAFFC.2025.3535542.
>
> > **For different tasks/datasets, it doesn't seem that the proposed methods were compared with SOTAs. Why only baselines were compared? The authors should compare their results with SOTA performances of the datasets, e.g., Seed, Faced. E.g., the results from comparison methods for VEP in table 3 are far away from SOTA performances reported in the literature. The gap between the proposed method and other methods is too good to believe. More clarification is needed.**
>
> We actually compared various SOTA methods, as also acknowledged by reviewer HQzc.
> Since HEEGNet is proposed for general EEG decoding, we focus on comparison with classical and SOTA models also proposed for general EEG in the main text of the manuscript.
> SOTA methods proposed for VEP and emotion results are summarized in the Appendix.
> The performance gaps are because all methods are evaluated under the stricter multi-source multi-target SFUDA setting, which is substantially more challenging than the supervised or mixed-subject settings.
> As shown in Tables 8 and 9 (per-dataset results), many methods exhibit highly competitive performance on certain datasets.
> Requested by other reviewers, we moreover include foundation models and manifold-decoding baselines in the revised manuscript.
>
>
> > **-- It is also not clear whether the methods are tested in the cross-dataset setting. E.g, for the Emotion recognition task, are the Seed and Faced datasets combined or tested separately?**
>
> Thank you for your question.
> In this work, we consider cross-subject and cross-session domain adaptation within each dataset, so we do not combine any datasets.
> We clarify it in the experiment section explicitly in the revised manuscript.
>
> > **Minor comment: Abstract is a bit misleading. Reviewers would think this is the first work on exploring hyperbolic embeddings for EEG.**
>
> Thank you for your suggestions. We have revised the abstract accordingly.

---

> ### Author Response · Authors · 2025-11-19
> **Response to Official Review by Reviewer Qng1 - questions**
>
> ***Questions***:
>
> > **The authors should carefully clarify their novelty.**
>
> Thank you for your question.
>
> In our work, we utilize hyperbolic geometry to tackle the unsupervised domain adaptation problem in general EEG signals.
> We first empirically demonstrate that EEG data exhibits hyperbolicity, and that hyperbolic embeddings improve cross-domain generalization and class separability.
> We then propose the hybrid model design of Euclidean encoders and hyperbolic neural networks to balance feature encoding and hierarchical structural representation and obtain high-quality EEG embeddings.
> The quality of embeddings plays a crucial role in domain adaptation; better embeddings generally lead to improved adaptation performance.
> We further propose a domain adaptation strategy, DSMDBN, that combines moments alignment and feature distribution alignment, achieving state-of-the-art results across multiple tasks.
> Through ablation studies, we demonstrate the effectiveness of hyperbolic embeddings (Table 2) and our novel DSMDBN adaptation strategy (Table 4).
>
> > **The performance comparison part need to be significantly improved and justified.**
>
> Thank you for your question.
> In this work, we consider cross-domain adaptation within each dataset.
> We evaluate HEEGNet and all baselines under a strict multi-source, multi-target, source-free unsupervised domain adaptation setting (Section 2.2).
> Since HEEGNet is proposed for general EEG decoding, we focus on comparisons of classical and recent SOTA models also intended for general EEG decoding in the main text of the manuscript.
> SOTA models proposed for VEP and emotion recognition, along with per‑dataset results, are summarized in the Appendix.
> In the revised manuscript, we also include comparisons with foundation models and manifold‑decoding baselines.
> Across all tasks and datasets, HEEGNet achieves competitive performance.
> Through ablation studies, we demonstrate that HEEGNet benefits from both hyperbolic embeddings (Table 2) and our novel DSMDBN adaptation strategy (Table 4).

---

> ### Comment · Reviewer_Qng1 · 2025-11-28
> **HEEGNet: Hyperbolic Embeddings for EEG**
>
> The authors clarified the motivation and novelty of the proposed method. Thanks.
>
> However, it is still not clear why the proposed HEEGNet is not compared with other hyperbolic embeddings based methods. The authors explained why Jing Chang 2025 can't be compared, but is there any hyperbolic embeddings based method for comparison?
>
> Also, almost all baselines are pretty much before 2023, it's not convincing that they are SOTAs.  For emotion recognition,  EmT is from 2025, but its performance is very bad (Table 9); similarly the 2 foundation models from 2024 provide very bad performance (e.g., in Table 9, LaBraM gives 14.9 ± 3.3 while DSMDBN+EA gives 89.7 ± 11.2 for Faced ). It's unexpected to see so poor performances from the comparison methods. It raises the question whether such methods are most suitable for comparison.
>
> The authors didn't address my comments:
> -- why not compare with SOTAs for each dataset? (The authors argued the 'general' aspect. However if a general method doesn't yield comparable performance when compared with SOTAs for specific problems, people will not use general methods.)
> -- how come the methods used for comparison in the tables perform so bad? Are they suitable methods?

---

> > ### Author Response · Authors · 2025-12-03
> > **Response to Reviewer Qng1 questions**
> >
> > Thank you for acknowledging our novelty and for further questions.
> > Please find our detailed responses to the questions below.
> >
> > > **However, it is still not clear why the proposed HEEGNet is not compared with other hyperbolic embeddings based methods. The authors explained why Jing Chang 2025 can't be compared, but is there any hyperbolic embeddings based method for comparison?**
> >
> > To the best of our knowledge, we are the first to utilize hyperbolic geometry for EEG classification. To provide a comprehensive comparison, we also include other geometry-based decoding approaches, such as Grassmann manifold and SPD manifold methods, in Table 5 to benchmark against the hyperbolic manifold.
> >
> > > **Also, almost all baselines are pretty much before 2023, it's not convincing that they are SOTAs. For emotion recognition, EmT is from 2025, but its performance is very bad (Table 9); similarly the 2 foundation models from 2024 provide very bad performance (e.g., in Table 9, LaBraM gives 14.9 ± 3.3 while DSMDBN+EA gives 89.7 ± 11.2 for Faced ). It's unexpected to see so poor performances from the comparison methods. It raises the question whether such methods are most suitable for comparison.**
> >
> > As acknowledged by Reviewer HQzc and Reviewer b7a4, we have compared our method against a broad range of state-of-the-art approaches.
> > Regarding the unexpectedly low performance of LaBraM, we initially followed the configuration without specifying input channels.
> > It appears that omitting this step can lead to a substantial performance drop for LaBraM.
> > We have now added the channel definitions and have updated the results accordingly.
> >
> > > **The authors didn't address my comments: -- why not compare with SOTAs for each dataset? (The authors argued the 'general' aspect. However if a general method doesn't yield comparable performance when compared with SOTAs for specific problems, people will not use general methods.) -- how come the methods used for comparison in the tables perform so bad? Are they suitable methods?**
> >
> > Our goal is to develop a generalizable unsupervised domain adaptation framework that operates without any target labels—a requirement that is crucial for real-world EEG applications.
> > In many practical and clinical scenarios, researchers have access to EEG recordings from a variety of tasks, and thus a general approach capable of handling distribution shifts is needed. This challenge has been widely recognized as a major open problem in EEG decoding (Fairclough, +2020).
> >
> > HEEGNet is proposed for general EEG decoding; we focus on comparison with classical and SOTA models also proposed for general EEG in the main text of the manuscript.
> > SOTA models proposed for specific tasks are summarized in the Appendix.
> > The observed performance gaps arise because all methods are evaluated under the stricter multi-source, multi-target SFUDA setting, which is substantially more challenging than supervised or mixed-subject configurations. As shown in Tables 8 and 9 (per-dataset results), many methods do achieve competitive performance on specific datasets.
> > Moreover, when these methods are combined with SFUDA baselines—Euclidean Alignment or Spatio-Temporal Monge Alignment (STMA)—their performance further improves.
> >
> > *Reference*
> >
> > Stephen H Fairclough and Fabien Lotte. Grand challenges in neurotechnology and system neuroer-
> > gonomics. Frontiers in Neuroergonomics, 1:602504, 2020. doi: 10.3389/fnrgo.2020.602504.

---

### Official Review · Reviewer_VB3w · 2025-11-01

**Soundness:** 2
**Presentation:** 4
**Contribution:** 3
**Rating:** 6
**Confidence:** 4

**Summary:**

The paper proposes a  hyperbolic architecture based on EEGNet for EEG‑based task with a novel domain‑shift alignment method in hyperbolic space.

**Strengths:**

The two-step alignment strategy, DSMDBN, operates in hyperbolic space: it first aligns source and target domain distributions by matching their first and second moments using a hyperbolic BatchNorm module, and then further aligns the moment-normalized features to a standard hyperbolic Gaussian via the Horospherical Sliced-Wasserstein (HHSW) loss. This approach is well-motivated and offers a moderately novel contribution to the domain adaptation literature.

**Weaknesses:**

1. Hyperbolicity evidence may reflect model bias:

The observed hyperbolicity is derived from learned embeddings rather than the intrinsic geometry of the EEG signals. Therfore, the results may reflect model-induced structure rather than inherent data-level hyperbolic properties.

2. Numerical stabilization not addressed:

The paper does not discuss numerical stabilization techniques essential for reliable hyperbolic training, such as gradient clipping, norm normalization, or overflow prevention during Lorentzian operations.

3. Missing comparisons with recent foundation models

This work lacks comparison with recent high-impact foundation models such as BIOT (NeurIPS' 2023) [1], LaBraM (ICLR' 2024) [2], and CBraMod (ICLR' 2025) [3]. These models are widely cited, have open-source code and model weights provided, and achieve state-of-the-art results across EEG decoding tasks.


Refs:

[1] Yang, Chaoqi, M. Westover, and Jimeng Sun. "Biot: Biosignal transformer for cross-data learning in the wild." Advances in Neural Information Processing Systems 36 (2023): 78240-78260.

[2] Jiang, Weibang, Liming Zhao, and Bao-liang Lu. "Large Brain Model for Learning Generic Representations with Tremendous EEG Data in BCI." The Twelfth International Conference on Learning Representations.

[3 Wang, Jiquan, et al. "CBraMod: A Criss-Cross Brain Foundation Model for EEG Decoding." The Thirteenth International Conference on Learning Representations.

**Questions:**

1. How is the curvature K selected?

---

> ### Author Response · Authors · 2025-11-19
> **Response to Official Review by Reviewer VB3w - weaknesses & questions**
>
> Thank you very much for your time to assess our submission and the provided feedback.
> For your convenience, we include copies of the revised manuscript [manuscript_revised.pdf](https://anonymous.4open.science/r/HEEGNet-F655/revised_version.pdf) and the submitted manuscript [manuscript_submitted.pdf](https://anonymous.4open.science/r/HEEGNet-F655/submitted_version.pdf).
>
> Please find our detailed responses to the weaknesses and questions below.
>
>
> ***Weaknesses***
>
> > **Hyperbolicity evidence may reflect model bias:
> The observed hyperbolicity is derived from learned embeddings rather than the intrinsic geometry of the EEG signals. Therfore, the results may reflect model-induced structure rather than inherent data-level hyperbolic properties.**
>
> Interesting point!
> We now include $\delta$-hyperbolicity analysis (Table 1) for raw EEG data, EEGNet-generated embeddings from intermediate layers (after the first two convolutional layers), and the classification space embeddings.
> We find that both raw EEG data and generated embeddings exhibit hierarchical structure, but the raw EEG data shows slightly stronger hyperbolicity.
> This indicates that EEGNet actually attenuates the hierarchicay, and is not suitable to capture the hierarchical structure.
>
> Nevertheless, our fully hyperbolic experiment (Appendix F) demonstrated that extracting a meaningful feature space with a Euclidean backbone is essential for effective downstream classification.
> This is becuase Euclidean convolutions are essential for modeling the temporal, spatial, and spectral structures of EEG signals; such operations cannot be replaced on the hyperbolic manifold.
> These findings support the rationale behind our hybrid architecture, which combines Euclidean encoders with hyperbolic neural networks.
>
>
> > **Numerical stabilization not addressed:
> The paper does not discuss numerical stabilization techniques essential for reliable hyperbolic training, such as gradient clipping, norm normalization, or overflow prevention during Lorentzian operations.**
>
> Thank you for pointing out this important detail.
> In our experiments, we applied norm normalization before projection onto the hyperbolic space.
> We have now included this information in the revised manuscript, in the experimental section.
>
> > **Missing comparisons with recent foundation models
> This work lacks comparison with recent high-impact foundation models such as BIOT (NeurIPS' 2023) [1], LaBraM (ICLR' 2024) [2], and CBraMod (ICLR' 2025) [3]. These models are widely cited, have open-source code and model weights provided, and achieve state-of-the-art results across EEG decoding tasks.**
>
> Thank you for your suggestions.
> We have now included LaBraM and CBraMod as baselines for motor imagery, VEP, emotion recognition, and iEEG tasks in Table 5, Table 8, Table 9, and Table 10 of the revised manuscript.
> Still, the results remain a gap to our proposed method, foundation models seems to easily overfit on relatively small datasets.
> Due to limited time, we are still running experiments on BIOT.
>
> ***Questions***
>
> > **How is the curvature K selected?**
>
> Thank you for your question.
> Following current literature (Bdeir+2024, ICLR; Mishne+2023, ICML), we set the curvature K to -1 by default.
> We included this information in the revised manuscript.
>
> *References*:
> - Ahmad Bdeir, Kristian Schwethelm, and Niels Landwehr. Fully hyperbolic convolutional neural networks for computer vision. In ICLR, 2024
>
> - Gal Mishne, Zhengchao Wan, Yusu Wang, and Sheng Yang. The numerical stability of hyperbolic representation learning. In International Conference on Machine Learning, pp. 24925-24949. PMLR, 2023.

---

### Official Review · Reviewer_b7a4 · 2025-11-01

**Soundness:** 3
**Presentation:** 3
**Contribution:** 3
**Rating:** 6
**Confidence:** 4

**Summary:**

This paper proposes HEEGNet, a hybrid hyperbolic embedding network designed to improve cross-domain generalization in EEG-based emotion recognition tasks. By jointly leveraging Euclidean and hyperbolic geometries, the model aims to capture both local and hierarchical relations within EEG representations. The proposed framework is evaluated on the SEED dataset and demonstrates competitive or superior performance compared to state-of-the-art methods. The manuscript is generally well-structured and technically sound, addressing an important challenge in EEG domain adaptation. However, several conceptual and methodological aspects require clarification, particularly regarding the motivation and interpretation of the hybrid embedding design and the underlying assumptions about EEG data structure.

**Strengths:**

1. The paper tackles an important issue—cross-domain generalization in EEG signals—where conventional Euclidean-based networks often fail to model hierarchical dependencies or inter-domain variations effectively.
2. The hybrid use of Euclidean and hyperbolic embeddings provides an interesting perspective, allowing the model to balance local feature encoding and hierarchical structural representation. This design is conceptually meaningful for EEG data, which may contain both flat temporal dynamics and hierarchical cognitive relations.
3. Experiments on SEED show consistent improvements over relevant baselines, suggesting that hyperbolic modeling can indeed improve domain transfer and robustness in EEG-based emotion recognition.

**Weaknesses:**

1. The manuscript claims that EEG, video, and language modalities can all be represented in hyperbolic space, but does not provide sufficient theoretical or empirical justification for why EEG data, in particular, exhibits hierarchical or tree-like properties that make hyperbolic geometry appropriate.
2. The rationale for combining Euclidean and hyperbolic representations is underdeveloped. It remains unclear what complementary features each space captures and how their joint use specifically benefits cross-domain transfer.
3. Terms such as “neural information” lack precision, and the formulation of domain adaptation within the SEED dataset (e.g., session-wise vs. subject-wise domains) should be clarified. These ambiguities make it difficult to interpret the mechanism behind improved generalization.
4. While t-SNE visualizations are provided, they reveal sub-clustering within the same class. The manuscript does not discuss why this occurs or whether it reflects subject-level variation, embedding instability, or meaningful substructure. Additional visualization or embedding analysis could better support the claimed advantages of hyperbolic modeling.

**Questions:**

1. Please clarify whether EEG, video, and natural language modalities share structural similarities that justify their representation in hyperbolic space. How do the manifold assumptions differ across these modalities?
2. Could you elaborate on the motivation for integrating Euclidean and hyperbolic embeddings? What distinct geometric properties does each space capture in EEG data?
3. The term “neural information” (Line 93, Page 2) is vague. Please specify the type of neural features it refers to—temporal, spectral, or spatial—and how they are encoded in your model.
4. In the SEED dataset experiments, how are domains defined? Are sessions or subjects treated as distinct domains, and how is cross-domain generalization evaluated?
5. It is recommended to visualize intermediate representations from both Euclidean and hyperbolic branches to more directly show their complementary effects and the benefits of the hyperbolic layer.
6. In Fig. 2(d), samples of the same class form multiple clusters. Please explain this behavior—does it arise from subject variability, intra-class diversity, or properties of hyperbolic projection?

---

> ### Author Response · Authors · 2025-11-19
> **Response [1/2] to Official Review by Reviewer b7a4 - weaknesses**
>
> Thank you very much for your time to assess our submission and the provided feedback.
> For your convenience, we include copies of the revised manuscript [manuscript_revised.pdf](https://anonymous.4open.science/r/HEEGNet-F655/revised_version.pdf) and the submitted manuscript [manuscript_submitted.pdf](https://anonymous.4open.science/r/HEEGNet-F655/submitted_version.pdf).
>
> Please find our detailed responses to the weaknesses and questions below.
>
> > **The manuscript claims that EEG, video, and language modalities can all be represented in hyperbolic space, but does not provide sufficient theoretical or empirical justification for why EEG data, in particular, exhibits hierarchical or tree-like properties that make hyperbolic geometry appropriate.**
>
> Thank you for raising this point.
> Our use of hyperbolic space for EEG is motivated by both theoretical and empirical evidence.
> From a theoretical perspective, the brain is widely understood as a hierarchical functional network, and neuroscientific studies have shown that EEG signals can reflect hierarchical cognitive processing [1][2][3].
> For example, in visual processing, lower cortical areas detect basic features, which higher cortical areas progressively refine into a global representation.
> These findings provide a well-established basis for considering hierarchical structure in EEG representations.
>
> For empirical justification, we employ $\delta$-hyperbolicity [4][5], a well-established metric to determine hierarchical and tree-like structures (See an explanation in Appendix C.8).
> In our experiments (Table 1), we compute $\delta$-hyperbolicity of raw EEG data, EEGNet-generated embeddings from intermediate layers (after the first two convolutional layers), and the classification space embeddings.
> We observed that both raw EEG data and EEGNet generated embeddings exhibit hierarchical structures.
>
> *References*
>
> [1] Elliot Collins, Amanda K Robinson, and Marlene Behrmann. Distinct neural processes for the perception of familiar versus unfamiliar faces along the visual hierarchy revealed by eeg. NeuroImage, 181:120-131, 2018. doi: 10.1016/j.neuroimage.2018.06.080.
>
> [2] Sai Sun, Hongbo Yu, Rongjun Yu, and Shuo Wang. Functional connectivity between the amygdala and prefrontal cortex underlies processing of emotion ambiguity. Translational psychiatry, 13(1): 334, 2023. doi: 10.1038/541398-023-02625-W.
>
> [3] William Turner, Tessel Blom, and Hinze Hogendoorn. Visual information is predictively encoded in occipital alpha/low-beta oscillations. Journal of Neuroscience, 43(30):5537-5545, 2023. doi: 10.1523/JNEUROSCI.0135-23.2023.
>
> [4] Valentin Khrulkov, Leyla Mirvakhabova, Evgeniya Ustinova, Ivan Oseledets, and Victor Lempitsky. Hyperbolic image embeddings. In Proceedings of the IEEE/CVF conference on computer vision and pattern recognition, pp. 6418-6428, 2020.
>
> [5] Ahmad Bdeir, Kristian Schwethelm, and Niels Landwehr. Fully hyperbolic convolutional neural networks for computer vision. In ICLR, 2024

---

> ### Author Response · Authors · 2025-11-19
> **Response [1/2] to Official Review by Reviewer b7a4 - weaknesses**
>
> > **The rationale for combining Euclidean and hyperbolic representations is underdeveloped. It remains unclear what complementary features each space captures and how their joint use specifically benefits cross-domain transfer..**
>
> We appreciate the reviewer comments.
> The rationale for combining Euclidean and hyperbolic representations is motivated by the complementary strengths of the two spaces.
> Euclidean convolutions are essential for modeling the temporal, spatial, and spectral structures of EEG signals.
> Such operations do not preserve physical properties in hyperbolic space, where dimensions are intrinsically coupled, making standard convolutional operations along that dimension no longer applicable.
> Thus, Euclidean encoding captures local signal structures that cannot be replaced on the hyperbolic manifold.
>
> In our design, Euclidean encoding captures signal structures that are difficult to represent effectively in hyperbolic space, while the hyperbolic component models the hierarchical relationships in the data.
> This design allows us to get high quality embeddings, the quality of embeddings plays a crucial role in domain adaptation; better embeddings generally lead to improved adaptation performance.
>
> This rationale is supported by our experiment in Appendix F. When we replaced the Euclidean convolutional layers in EEGNet with hyperbolic variants, performance dropped significantly, indicating that a fully hyperbolic architecture cannot adequately capture the local EEG signal characteristics required for effective decoding.
>
> > **Terms such as “neural information” lack precision, and the formulation of domain adaptation within the SEED dataset (e.g., session-wise vs. subject-wise domains) should be clarified. These ambiguities make it difficult to interpret the mechanism behind improved generalization**
>
> We appreciate the your comment and have carefully checked the manuscript, we could not identify the word "neural information" in our manuscript.
>
> Regarding the formulation of domains, for all datasets, we treat sessions as domains, which means we calculate domain statistics at thesession level.
> We consider the cross-subject adaptation and treat the subject as the grouping variable and use either a leave-one-group-out (group number $\leq$10) or a 10-fold leave-groups-out cross-validation scheme to fit and evaluate models.
>
> > **While t-SNE visualizations are provided, they reveal sub-clustering within the same class. The manuscript does not discuss why this occurs or whether it reflects subject-level variation, embedding instability, or meaningful substructure. Additional visualization or embedding analysis could better support the claimed advantages of hyperbolic modeling.**
>
> Thank you for pointing this out.
> In our t-SNE visualizations, we intentionally used the same subject and the same session across different training configurations — Euclidean embeddings, hyperbolic multinomial logistic regression, moments alignment, and DSMDBN (our proposed method).
> This controlled setup allows us to directly compare the relative advantages of hyperbolic geometry and our adaptation modules.
>
> The observed sub-clustering within a class does not reflect subject-level variation, since the subject and session are fixed.
> Instead, it indicates that the multi-modal structure of the class-conditional distribution is preserved in the embeddings, which is not necessarily a bad thing.

---

> ### Author Response · Authors · 2025-11-19
> **Response to Official Review by Reviewer b7a4 - questions**
>
> > **Please clarify whether EEG, video, and natural language modalities share structural similarities that justify their representation in hyperbolic space. How do the manifold assumptions differ across these modalities?**
>
> Thank you for your question.
> Our use of hyperbolic space is grounded in the hierarchical structure of the EEG, which has been demonstrated in neuroscience studies [3][4].
> Prior work [1][2] in hyperbolic manifolds further shows that hyperbolic geometry is well-suited for any data whose latent structure is tree-like or hierarchical, regardless of modality.
>
> *References*
>
> [1] Yang, Menglin, et al. "Hyperbolic representation learning: Revisiting and advancing." International Conference on Machine Learning. PMLR, 2023.
>
> [2] Peng, Wei, et al. "Hyperbolic deep neural networks: A survey." IEEE Transactions on pattern analysis and machine intelligence 44.12 (2021): 10023-10044.
>
> [3] Elliot Collins, Amanda K Robinson, and Marlene Behrmann. Distinct neural processes for the perception of familiar versus unfamiliar faces along the visual hierarchy revealed by eeg. NeuroImage, 181:120-131, 2018. doi: 10.1016/j.neuroimage.2018.06.080.
>
> [4] Sai Sun, Hongbo Yu, Rongjun Yu, and Shuo Wang. Functional connectivity between the amygdala and prefrontal cortex underlies processing of emotion ambiguity. Translational psychiatry, 13(1): 334, 2023. doi: 10.1038/541398-023-02625-W.
>
> > **Could you elaborate on the motivation for integrating Euclidean and hyperbolic embeddings? What distinct geometric properties does each space capture in EEG data?**
>
> Thank you for your question.
> Euclidean encoders are well-established for temporal, spatial, and spectral structures in EEG.
> Such operations do not preserve physical properties in hyperbolic space, where dimensions are intrinsically coupled.
> Hyperbolic embedding helps to preserve and refine the hierarchical
> structure in hyperbolic space.
>
> In our experiments, we first demonstrate that raw EEG data and Euclidean encoders generated embeddings exhibit hierarchical structures;
> Replacing the EEGNet multinomial logistic regression with a hyperbolic variant improved generalization (Table 2) and class separability  (Figure 2a and 2b).
> Our experiment in Appendix F further supports this design, where we observed a huge performance drop for fully hyperbolic neural networks.
>
>
> > **The term “neural information” (Line 93, Page 2) is vague. Please specify the type of neural features it refers to—temporal, spectral, or spatial—and how they are encoded in your model.**
>
> We could not identify the word "neural information" in our manuscript.
> We sequentially adopt the three convolutional layers from EEGNet: temporal convolution to learn frequency-specific filters, depthwise spatial convolution to capture electrode-wise patterns, and a second depthwise temporal convolution to summarize information across time.
> These well-established operations provide spectral-spatial-temporal feature maps.
>
> > **In the SEED dataset experiments, how are domains defined? Are sessions or subjects treated as distinct domains, and how is cross-domain generalization evaluated?**
>
> Thank you for your question.
> We treat sessions as domains, which means we calculate domain statistics at the session level.
> We consider the cross-subject adaptation and treat the subject as the grouping variable and use either a leave-one-group-out (group number $\leq$10) or a 10-fold leave-groups-out cross-validation scheme to fit and evaluate models.
>
>
> > **It is recommended to visualize intermediate representations from both Euclidean and hyperbolic branches to more directly show their complementary effects and the benefits of the hyperbolic layer.**
>
> Thank you for your recommendation.
> We provide t-SNE visualizations in Figures 2a-2d for demonstration.
> t-SNE visualizations of the same subject and session under different training configurations—Euclidean embeddings (2a), hyperbolic multinomial logistic regression (2b), moment alignment (2c), and the proposed DSMDBN (2d)—show that, with the same encoder structure, hyperbolic representations and our domain adaptation strategy systematically improve class separability.
>
> > **In Fig. 2(d), samples of the same class form multiple clusters. Please explain this behavior—does it arise from subject variability, intra-class diversity, or properties of hyperbolic projection?.**
>
> Thank you for your question
> We intentionally visualize the same subject and same session under different training configurations—Euclidean embeddings, hyperbolic multinomial logistic regression, moment alignment, and DSMDBN (proposed).
> The observed sub-clustering within a class does not reflect subject-level variation, since the subject and session are fixed.
> The sub-clusters indicate that the multi-modal structure of the class conditional was preserved, which is not necessarily a bad thing.

---

> > ### Comment · Reviewer_b7a4 · 2025-11-28
> > **Revised Review After Authors’ Response**
> >
> > I appreciate the authors’ detailed and thoughtful rebuttal. My earlier concerns—particularly regarding the justification of hyperbolic representations for EEG, the empirical evidence supporting hierarchical structure in neural signals, and the stability/generalization improvements under domain shifts—were all addressed satisfactorily.
> > The authors have provided clearer motivation for modeling EEG in hyperbolic space and strengthened the empirical justification by showing explicit evidence of hyperbolicity in EEG data. Additionally, the extended experimental analyses across multiple EEG benchmarks (visual evoked potentials, emotion recognition, and intracranial EEG) more convincingly demonstrate the generalization benefits.
> > Overall, I now believe that the proposed HEEGNet represents a meaningful and well-motivated contribution to EEG representation learning and cross-domain robustness. I consider the paper to meet the standard of ICLR, and I will raise my score accordingly.

---

### Author Response · Authors · 2025-11-19
**Global Response**

Dear reviewers, we appreciate your precious the time in reviewing our paper and provide valuable comments.
We uploaded the revised manuscript and will provide the point-by-point responses to your comments shortly.
All reviewer comments have been carefully considered, and we have made every effort to address them comprehensively.
We hope the revised manuscript meets your high standards.

For your convenience, we include copies of  the revised manuscript [manuscript_revised.pdf](https://anonymous.4open.science/r/HEEGNet-F655/revised_version.pdf) and the originally submitted manuscript  [manuscript_submitted.pdf](https://anonymous.4open.science/r/HEEGNet-F655/submitted_version.pdf) in the anonymous repository:

Below, we respond to the main questions and concerns we received from reviewers

- Novelty of our work

In this study, we introduce HEEGNet, which utilizes hyperbolic geometry to tackle the unsupervised domain adaptation problem in EEG.
We first empirically demonstrate that EEG data exhibits hyperbolicity and that hyperbolic embeddings improve cross-domain generalization.
We propose a hybrid model combining Euclidean encoders and hyperbolic neural networks to balance feature encoding with hierarchical structural representation, producing high‑quality EEG embeddings, as better embeddings generally lead to improved generalization.  We further propose a domain adaptation strategy, DSMDBN, that combines moments alignment and feature distribution alignment, achieving state-of-the-art results across multiple tasks.

- Motivation for hyperbolic geometry in EEG

Our motivation is grounded in neuroscientific evidence showing that EEG signals can reflect hierarchical cognitive processing [1][2][3].
These findings provide a well-established basis for considering hierarchical structure in EEG representations.

For empirical justification, we employ $\delta$-hyperbolicity [4][5], a well-established metric to determine hierarchical and tree-like structures (See an explanation in Appendix C.8).
In our experiments (Table 1), we compute $\delta$-hyperbolicity of raw EEG data, EEGNet-generated embeddings from intermediate layers (after the first two convolutional layers), and the classification space embeddings.
We observed that both raw EEG data and generated embeddings exhibit hierarchical structures.

Prior work [6][7]  shows that hyperbolic geometry is well-suited for any data whose latent structure is tree-like or hierarchical, regardless of modality.
Our results confirm that hyperbolic embeddings improve cross-domain generalization (Table 2) and enhance class separability (Figure 2), providing strong empirical support for using hyperbolic geometry in EEG.

- Comparison of experimental results

In this work, we consider cross-domain adaptation within each dataset.
We evaluate HEEGNet and all baselines under the multi-source, multi-target, source-free unsupervised domain adaptation setting (Section 2.2).
Since HEEGNet is proposed for general EEG decoding, we focus on comparisons of classical and SOTA models also intended for general EEG decoding in the main text of the manuscript.
SOTA models proposed for VEP and emotion recognition, along with per‑dataset results, are summarized in the Appendix.
In the revised manuscript, we also include comparisons with foundation models and manifold‑decoding baselines.
Across all tasks and datasets, HEEGNet achieves competitive performance.
Through ablation studies, we demonstrate that HEEGNet benefits from both hyperbolic embeddings (Table 2) and our novel DSMDBN adaptation strategy (Table 4).

*References*

[1] Elliot Collins, Amanda K Robinson, and Marlene Behrmann. Distinct neural processes for the perception of familiar versus unfamiliar faces along the visual hierarchy revealed by eeg. NeuroImage, 181:120-131, 2018. doi: 10.1016/j.neuroimage.2018.06.080.

[2] Sai Sun, Hongbo Yu, Rongjun Yu, and Shuo Wang. Functional connectivity between the amygdala and prefrontal cortex underlies processing of emotion ambiguity. Translational psychiatry, 13(1): 334, 2023. doi: 10.1038/541398-023-02625-W.

[3] William Turner, Tessel Blom, and Hinze Hogendoorn. Visual information is predictively encoded in occipital alpha/low-beta oscillations. Journal of Neuroscience, 43(30):5537-5545, 2023. doi: 10.1523/JNEUROSCI.0135-23.2023.

[4] Valentin Khrulkov, Leyla Mirvakhabova, Evgeniya Ustinova, Ivan Oseledets, and Victor Lempitsky. Hyperbolic image embeddings. In Proceedings of the IEEE/CVF conference on computer vision and pattern recognition, 2020.

[5] Ahmad Bdeir, Kristian Schwethelm, and Niels Landwehr. Fully hyperbolic convolutional neural networks for computer vision. In ICLR, 2024

[6] Yang, Menglin, et al. "Hyperbolic representation learning: Revisiting and advancing." International Conference on Machine Learning. PMLR, 2023.

[7] Peng, Wei, et al. "Hyperbolic deep neural networks: A survey." IEEE Transactions on pattern analysis and machine intelligence 44.12 (2021): 10023-10044.

---

### Author Response · Authors · 2025-11-27
**We look forward to your replies.**

Dear reviewers,

As the discussion phase is approaching its end, we would like to kindly follow up regarding our submission.
We would be grateful to know if our clarifications have addressed your concerns.
We would be happy to discuss any additional points the reviewer may have.

We look forward to your replies
With best regards,
Submission312 Authors

---

### Author Response · Authors · 2025-12-03
**Overall Summary to AC**

Dear Area Chair,

We sincerely thank you for your additional time and effort in reevaluating the discussion and our submission.

Overall, we received a 6, 6, 4, 6 score before rebuttal, where reviewer b7a4  promised to increase their score from 6 during the discussion period.

In this work, we tackle the unsupervised domain adaptation problem in EEG with the following two main contributions:
- Introducing hyperbolic geometry for EEG classification
- Proposing a novel domain adaptation framework

These contributions were acknowledged by reviewers b7a4, VB3w, and HQzc before the rebuttal, and by reviewer Qng1 after the rebuttal.
Reviewer b7a4 noted that hyperbolic geometry to represent EEG is well-motivated and empirically justified after discussion, and reviewer HQzc favored its technical novelty.
Reviewer VB3w noted that our framework is “well-motivated and a novel contribution to domain adaptation literature.”


Specifically, we first empirically demonstrate that EEG data exhibits hyperbolicity and that hyperbolic embeddings improve cross-domain generalization.
We then propose a domain adaptation strategy, DSMDBN, that operates in hyperbolic space and combines moments alignment and feature distribution alignment, achieving state-of-the-art results across multiple tasks.


Below is a summary of the rebuttal.

---
**Reviewer b7a4: 6 (before rebuttal) $\rightarrow$ "explicitly note to raise the score" (after rebuttal)**
- Main concerns: justification of hyperbolic representations for EEG; rationale for hybrid design; visualization explainability; clarity of the experiment settings.
- Our rebuttal: provided empirical evidence supporting hierarchical structure in EEG (Table 1); conducted an ablation study to demonstrate the necessity of the hybrid design (Table 11); clarified and explained the visualization; clarified the source-free unsupervised domain adaptation experiment settings.
- Reviewer stance: **Explicitly note to raise the score.** All concerns were addressed in revision.

---

**Reviewer VB3w: 6 (before rebuttal)**
- Main concerns: justification of hyperbolic representations for EEG; implementation details of hyperbolic geometry; experiment comparison.
- Our rebuttal: provided empirical evidence supporting hierarchical structure in EEG (Table 1); added implementation details in section 4.2; added foundation model and manifold-based decoding baselines.
- Reviewer stance: No follow-up comment was posted before the system rollback.
However, all questions and concerns were fully answered in our response.
Concerns about the motivation for hyperbolic geometry are considered to have been addressed by Reviewer b7a4.

---

**Reviewer Qng1: 4 (before rebuttal)**
- Main concerns: clarity of the novelty; clarity of the experiment comparison and setting.
- Our rebuttal: clarified our novelty; explained the source-free unsupervised domain adaptation experiment settings; added foundation model and manifold-based decoding baselines.
- Reviewer stance: acknowledged our novelty; no follow-up comment was posted.
Concerns about the experiment comparison with state-of-the-art methods, while our experiment comparisons are noted to be sufficient by Reviewer b7a4 and HQzc.
However, all questions and concerns were fully answered in our response.


---
**Reviewer HQzc: 6 (before rebuttal)**
- Main concerns: justification of hyperbolic representations for EEG; computation cost of hyperbolic operation; domain adaptation generalizable ability; clarity of the experiment settings; asked for a discussion section
- Our rebuttal: provided empirical evidence supporting hierarchical structure in EEG (Table 1); added computational cost experiment (Table 7); provided new experiment result; clarified the source-free unsupervised domain adaptation experiment settings; added a discussion section.
- Reviewer stance: No follow-up comment was posted before the system rollback; however, all questions and concerns were fully answered in our response, and concerns about the motivation for hyperbolic geometry are considered to have been addressed by Reviewer b7a4.

---

### Meta-Review · Area_Chair_jMwb · 2026-01-08

**Summary:**

The paper proposes a hyperbolic-geometry variant of EEGNet called HEEGNet for source-free unsupervised domain adaptation in EEG decoding, motivated by hierarchical structure in neural signals and supported by empirical hyperbolicity analyses. The key decision-driving concern is (1) whether empirical gains are sufficiently justified and fairly benchmarked; (2) what aspects of the method drive improvements.

**Reviewer Concerns:**

Most concerns are addressed in rebuttal: (1) Motivation or novelty clarification relative to prior works; (2) Justification of hyperbolic representations and evidence of EEG hyperbolicity; (3) Experimental setting clarification and more comparisons and ablations; (4) hyperbolicity evidence may reflect learned embedding bias rather than data-level structure; (5) Numerical stabilization and hyperbolic training reliability (e.g., clipping/overflow prevention).

**Reviewer Scores:**

- b7a4: likely 6  to 7.
- VB3w: likely stays 6.
- HQzc: likely stays 6.
- Qng1: likely 4 to 6.

---

### Decision · Program_Chairs · 2026-01-26

Accept (Poster)